# iU-ExM: nanoscopy of organelles and tissues with iterative ultrastructure expansion microscopy

Vincent Louvel [1], Romuald Haase[2], Olivier Mercey[1], Marine H. Laporte[1], Thibaut Eloy[3], Étienne Baudrier [3], Denis Fortun[3], Dominique Soldati-Favre [2], Virginie Hamel [1] ✉ & Paul Guichard [1] ✉

Expansion microscopy (ExM) is a highly effective technique for super-resolution fluorescence microscopy that enables imaging of biological samples beyond the diffraction limit with conventional fluorescence microscopes. Despite the development of several enhanced protocols, ExM has not yet demonstrated the ability to achieve the precision of nanoscopy techniques such as Single Molecule Localization Microscopy (SMLM). Here, to address this limitation, we have developed an iterative ultrastructure expansion microscopy (iU-ExM) approach that achieves SMLM-level resolution. With iU-ExM, it is now possible to visualize the molecular architecture of gold-standard samples, such as the eight-fold symmetry of nuclear pores or the molecular organization of the conoid in Apicomplexa. With its wide-ranging applications, from isolated organelles to cells and tissue, iU-ExM opens new super-resolution avenues for scientists studying biological structures and functions.

Similar to conventional electron microscopy (EM) in the 20th century[1], the emergence of super-resolution microscopy (SRM) in the last two decades has revolutionized our vision of the molecular architecture of the cell, enabling the localization of proteins with resolution beyond the diffraction limit[2]. Of all existing methods, Single Molecule Localization Microscopy (SMLM) approaches, including PALM, STORM, PAINT, MINFLUX, have one of the highest resolving power with near-nanometre resolution[3], along with Structured Illumination Microscopy (SIM) and Stimulated Emission Depletion (STED). Although these technologies have made significant advancements in the study of sub-cellular organization, their use is limited to a small number of laboratories due to the need for state-of-the-art equipment and specialized expertise.

The development of expansion microscopy (ExM) in recent years enabled to democratize super-resolution by leveraging the isotropic expansion of the biological samples[4]. This innovative methodology based on a 4-fold increase of the distance between molecules, now allows obtaining a resolution in the order of 70 nm with a widefield or confocal microscope[4]. Since its original description, many variations of the expansion microscopy protocols have been developed to further increase the resolution[5]. There are three major approaches. The first one involves coupling expansion microscopy with super-resolution microscopes. In this case, it is possible to reveal the molecular architecture of sub-cellular ultrastructure with near electron microscopic precision such as the chirality of centrioles by coupling it with STED or dSTORM[6,7]. The second approach involves modifying the expanding polymer's chemistry to achieve a single expansion factor of around 10-fold for instance[8,9]. With this approach, nuclear pore complexes, which are circular structures of approximately 110 nm in diameter, appear as a ring-like structure, providing near-STED resolution[8]. Furthermore, by coupling 10× expansion with SRRF fluorescence fluctuation analysis[10], protein shapes can also be revealed. Finally, the third approach involves increasing the expansion factor by using multiple rounds of expansion, a method called iterative expansion microscopy, or iExM[11]. Among several iterative approaches, pan-ExM is a recent powerful one that combined iterative expansion and pan-labelling, demonstrating

[1]Department of Molecular and Cellular Biology, University of Geneva, Geneva, Switzerland. [2]Department of Microbiology and Molecular medicine, University of Geneva, Geneva, Switzerland. [3]ICube - UMR7357, CNRS, University of Strasbourg, Strasbourg, France. ✉e-mail: virginie.hamel@unige.ch; paul.guichard@unige.ch

electron microscopy-like resolution using measured expansion factors ranging from 13 to 21×[12].

However, despite the numerous developments made to enhance resolution, it appears that ExM alone, when used with a conventional fluorescence microscope, is still unable to achieve resolution comparable to that of SMLM[13]. Here, we push the resolution to that of SMLM by combining two approaches, Ultrastructure Expansion Microscopy (U-ExM)[6], known to best preserve the sub-cellular ultrastructure and reduce the linkage error and thus increase the labelling precision[14], with iterative expansion microscopy similar to iExM or pan-ExM methods[11,12]. Using nuclear pores, *Toxoplasma gondii* conoids, or other microtubule-based structures as molecular rulers, we optimized each step, particularly the homogenization and staining approach, to achieve an optimal molecular expansion of 16×. Our approach, called iterative U-ExM (iU-ExM), enhances the visualization of molecular details in ExM with a high level of precision. With iU-ExM, we are able to reveal the symmetry of nuclear pores, which had only been observed mainly using SMLM previously. Additionally, we demonstrate that iU-ExM is applicable to tissues and can visualize molecular periodicities along the connecting cilium of murine photoreceptor cells.

## Results

### Development of iterative U-ExM (iU-ExM)

Unlike the original ExM protocol, which employs proteinase K to fully digest the proteome and enable proper expansion[4], U-ExM, a variant of the MAP protocol[6,15], utilizes temperature denaturation (a process also called homogenization) at 95 °C to preserve the proteome while disrupting protein interactions prior to acrylamide/sodium acrylate gel expansion, thereby allowing for homogeneous isotropic expansion[6]. Furthermore, to best preserve subcellular architecture, we have demonstrated that, compared to the MAP protocol, the absence of glutaraldehyde in cell fixation and the optimization of protein anchoring through the use of a combination of low concentrations of formaldehyde (FA; 0.7–1.4%) and acrylamide (AA; 1–2%) is essential. Through this optimization, we were able to show that organelles were isotropically expanded, revealing their molecular architecture[6]. These optimal anchoring steps are the foundation of the pan-ExM protocol that combined these conditions with the iExM approach to reach an expansion factor of 13–21×[12]. The principle of iExM resides in using a first cleavable gel by replacing the cross-linker N,N′-Methylenbis-(acrylamid) (Bis-acrylamide) by N,N′-(1,2-Dihydroxy-1,2-ethanediyl) bis(acrylamide), called DHEBA, allowing a second round of expansion after re-embedding in a second uncleavable gel (Bis-acrylamide)[11]. By combining these approaches with a Pan-labelling using Hydroxysuccinimide (NHS) ester-dye conjugates that react with primary amines on proteins, pan-ExM protocol can reveal the cellular nanoarchitecture by standard light microscopy at a similar level to cellular electron microscopy[12,16]. However, in our hands, we found that immunostaining of certain organelles with the pan-ExM protocol, such as the centrioles using anti-tubulin antibodies, did not work, preventing us from measuring molecular isotropicity using this molecular ruler. We hypothesize that this limitation could come from a key distinction between pan-ExM and U-ExM during the denaturation step: while pan-ExM utilizes 73 °C denaturation during 1 h, U-ExM relies on a temperature of 95 °C during 1 h 30. We therefore tested this factor to image centrioles using anti-tubulin antibodies. We initially tested whether the DHEBA gel was compatible with a denaturation step at 95 °C but found that the DHEBA gel melted at this temperature, probably due to cross-linker cleavage, and thus decided to probe its solidity at lower temperatures. We have found that the gel can withstand 85 °C, approaching the denaturing conditions of the U-ExM. We thus tested 4 conditions, BIS-acrylamide based gel in U-ExM at 95 °C and 73 °C, and DHEBA based gel at 73 °C (pan-ExM condition) and 85 °C (Fig. 1a–d). Moreover, we omitted chemical fixation to avoid any effects on the molecular expansion factor[6], a condition known to

depolymerize cytoplasmic microtubules but retaining centrioles[17]. In these conditions, we found that centrioles were clearly detectable using anti-tubulin antibodies at 95 °C and 85 °C in both BIS and DHEBA based gels respectively (Fig. 1a, d). However, we could not detect centrioles after homogenization at 73 °C (Fig. 1b, c). To decipher between a problem of immunolabelling or centriole expansion, we co-stained cells with anti-tubulin antibodies and NHS ester ATTO 594 to visualise the entire proteome. We observed that centrioles were visible with antibodies and NHS ester ATTO 594 at 85 °C, but were only detectable with NHS-ester ATTO 594 at 73 °C (Fig. 1e). Furthermore, we found that the centrioles were poorly expanded (Fig. 1e). We concluded that rigid microtubule-based organelles might be difficult to expand at this temperature. To verify this hypothesis, we also tested these temperature conditions on the *T. gondii* parasite and found that the conoid, a tubulin fibers-based structure known to be isotropically expanded using U-ExM[18], was also not properly expanded after 1 h 73 °C denaturation (Fig. 1f). Similarly, doublecortin (DCX), a protein localizing at the level of the conoid[19], was also not properly detected[20] (Supplementary Fig. 1a, b). Based on these experiments, we concluded that the strength of homogenization at 73 °C was insufficient to achieve complete expansion. To confirm this, using human U2OS cells, we measured the expansion factor of the nucleus embedded in a DHEBA-based gel, treated at 73 °C for 1 h (pan-ExM condition), or at 85 °C for 1 h and 30 min (U-ExM-like condition). We found that the gel at 73 °C expanded by a factor of 4.6×, while the gel at 85 °C had a higher expansion factor of 5.8×, possibly due to the loss of some DHEBA crosslinkers that are cleaved at high temperature (Fig. 1g, Supplementary Fig. 1c, d). Using automatic binary segmentation of DAPI-stained nuclei, we determined the Nuclear Cross Section value (NCS), which corresponds to the square root of the nucleus area divided by the gel expansion factor (Fig. 1g, h). Through this analysis, compared with the non-expanded nucleus size, we observed that the nucleus homogenized at 73 °C did not expand as much as the gel, indicating that the expansion was likely not complete. However, as expected when using a higher denaturation temperature, the nuclei were better expanded at 85 °C, reaching complete expansion similar to U-ExM (Fig. 1g, h).

At the molecular level, we obtained similar results using the centriole as a ruler (Fig. 1i, j). Although the expansion appeared slightly smaller by a few nanometres in the 85 °C condition (Fig. 1i), the molecular isotropy, measured using the length-to-diameter ratio, was maintained (Fig. 1j) as observed in U-ExM. Since the centriole could not be stained for tubulin in the 73 °C condition, we proceeded with the iterative process and stained the centriole with NHS ester ATTO 594 to evaluate its expansion using homogenization at 85 °C (Fig. 1k). In agreement with our observations in the first gel, the iterative condition resulted in greater expansion at 85 °C, around 13× for the pan-ExM and 20× after denaturation at 85 °C (Supplementary Fig. 1e), producing a clearer image where the nine-fold symmetry of the proximal and distal centriole was clearly visible (Fig. 1k, l). Furthermore, thanks to the 1 h 30 85 °C denaturation, we could clearly observe that NHS ester ATTO 594 did not label the microtubules of the centriole, but seems to primarily mark the centriole's external surface, which may have biased the earlier measurements at 73 °C[12]. Therefore, we chose to adapt the denaturation step at 85 °C for further development of the iterative U-ExM, referred to as iU-ExM thereafter.

We next optimised the labelling step using antibodies against tubulin to visualize microtubules (Fig. 1m–o). We first tested the immunolabelling after iterative expansion (post-staining, Fig. 1m) but we found that the resulting signal was heterogenous, not existent, or very weak (Fig. 1n). We reasoned that this weak signal could be due to the density of the iterative gel, which may limit the diffusion of antibodies. We therefore tested whether an intermediate staining, i.e. in the first gel, would be retained in the second gel and improve the fluorescent signal (Fig.1m). In these conditions, the immunostaining

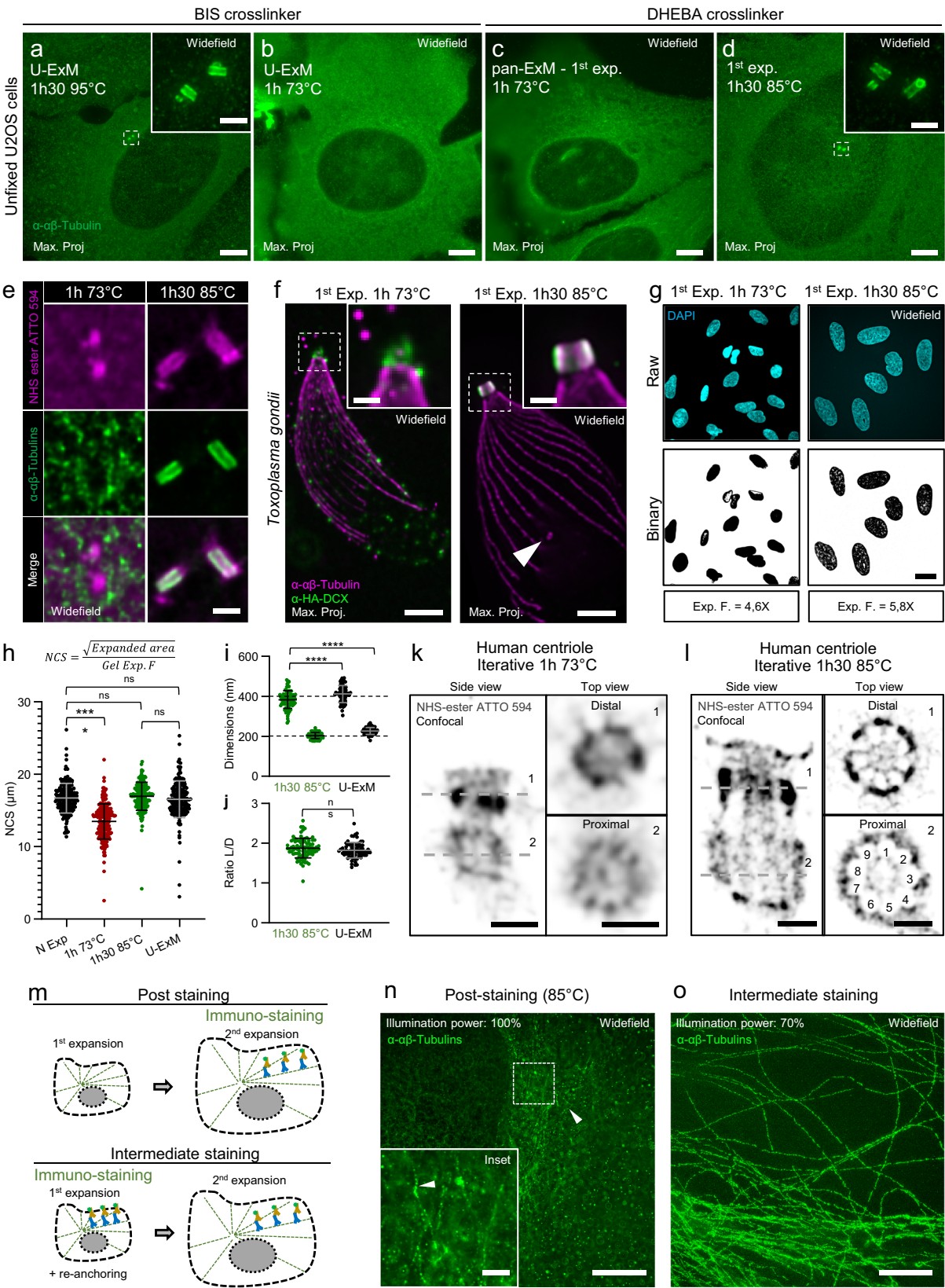

with the primary and secondary antibodies is done after expansion of the first DHEBA gel. The stained gel is then embedded in a neutral gel and the antibodies are thus anchored with a second AA/FA mix prior to the second expansion gel (Supplementary Fig. 1f). We found that this

approach led to a substantial increase in signal intensity and labelling coverage, as evidenced by the fluorescent signal of labelled microtubules (Fig. 1o). Therefore, we integrated this labelling approach with our iU-ExM methodology.

**Fig. 1 | iU-ExM development. a–d** Expanded U2OS unfixed cells stained for α and β tubulins. Scale bars: 10 µm non-corrected. **e** Centrioles stained for tubulin and NHS-ester ATTO 594 after the first expansion (unfixed U2OS). Scale bar: 300 nm corrected. **f** Expanded (first gel, unfixed) *T. gondii* tachyzoite conoid stained for tubulin and doublecortin (DCX-HA). Insets highlight the conoid structure. Scale bar: 400 nm, 300 nm (inset), both corrected. **g** Expanded nuclei from U2OS unfixed cells stained for DAPI. Scale bar: 100 µm non-corrected. **h** Quantification of the Nuclei Cross Section (NCS) in unfixed U2OS. N. Expanded (non-expanded): N = 215 (average ± standard error =16.7 ± 2 µm), 1 h 73 °C: N = 227 (average ± standard error = 13.5 ± 2.4 µm), 1 h30 85 °C: N = 189 (average ± standard error = 16.9 ± 1.9 µm), U-ExM: N = 218 (average ± standard error = 16.5 ± 2.5 µm) from 3 independent experiments. (**i**) Quantification of the centriole length and diameter corrected by the expansion factor of the gel in unfixed U2OS cells. 1 h 30 85 °C: N = 87 centrioles (average length ± standard error = 383.9 ± 43.9 nm, average diameter ± standard error = 204 ± 15.5 nm), 1 h30 95 °C (U-ExM): N = 73 centrioles (length: average ± standard error = 411.7 ± 43 nm, diameter: average ± standard error = 226.7 ± 17.3 nm) from 3 independent experiments. Measurements for the 73 °C condition were not possible due to incomplete centriole expansion and staining. **j** Centriole Length/Diameter ratio. 1 h30 85 °C: N = 87 centrioles (average ± standard error = 1.9 ± 0.2), 1 h30 95 °C (U-ExM): N = 73 centrioles (average ± standard error = 1.8 + 0.2) from 3 independent experiments. U2OS centrioles stained for NHS-ester ATTO 594 using either 73 °C (**k**) or 85 °C (iU-ExM) (**l**). Left: side view, right: top view. Scale bars: 100 nm corrected. **m** Scheme of post- and intermediate- staining strategies. PFA/GA fixed expanded U2OS cells with either post-staining (**n**) or intermediate staining (**o**). Scale bars: 50 µm non-corrected, 5 µm (inset). Non parametric Kruskal–Wallis test (**h**). One-way ANOVA, two-sided (**i**), unpaired two-sided t-test (**j**). ****p-value < 0.0001.

## The 8-fold symmetry of the nuclear pores revealed by iU-ExM

Next, we assessed the resolution of iU-ExM by testing whether it would enable the visualization of the nuclear pore complex (NPC). The NPC is a macromolecular assembly made in parts of two 8-fold symmetrical stacked rings used as a gold-standard reference structure to evaluate SRM performance that remains challenging to resolve using ExM[13]. We first tested fixation conditions as well as antibody labelling strategies using the validated NUP96-GFP cell line to visualize the NPC (Supplementary Fig. 2). NUP96 is a nucleoporin present in 32 copies in the NPC, with 4 copies per asymmetrical units of the rings, defined as 8 corners[13]. Consistent with the previous studies[13,21], we found that the optimal fixation condition for visualizing NPCs involved a pre-extraction-based protocol (see "Methods"), resulting in overall improved expansion compared to fixation with methanol (Supplementary Fig. 2a–d). Furthermore, we observed that utilizing a combination of antibodies targeting NUP96 and GFP yielded a stronger signal and enhanced antibody coverage of NPCs (Supplementary Fig. 2e–j).

Next, we assessed the capability of iU-ExM to reveal the 8-fold symmetry of NPC using widefield and confocal microscopes, and compared it with regular immunofluorescence, U-ExM, and STED imaging (Fig. 2a–d). By immunofluorescence without expansion, NPCs are observed as individual focal points (Fig. 2a). In U-ExM, nuclear pores begin to exhibit ring-like structures, albeit with a lower resolution compared to the clearly discernible ring shape in STED imaging (Fig. 2b, c). Using iU-ExM, we observed that NPCs were distinctly identifiable as separate rings using a regular widefield microscope (Fig. 2d). Additionally, we discovered that it was possible to detect the individual subunits comprising the 8-fold symmetry (Fig. 2d). Furthermore, by examining these structures from a side view in iU-ExM, we successfully identified the NUP96 localization within the nuclear and cytoplasmic rings. Collectively, these findings suggest that iU-ExM enables a resolution comparable to that achieved with SMLM. To further compare our results with quantitative super-resolution microscopy obtained through SMLM imaging on these cell lines[13], we proceeded to quantify the number of detectable NUP96 corners per NPC, 8 in total, as previously published[13,21,22] (Fig. 2e, f). We developed an automatic corner detection algorithm based on the proposed approach dedicated to localization coordinates in SMLM data[13] (see "Methods") (Fig. 2f). Through this analysis, we found that the observed NPC exhibited a distribution similar to previous SMLM studies[13,23], with a peak at 7 corners, which can be attributed to the labelling efficiency.

We then proceeded to test dual color staining, which often poses challenges in SMLM[5]. For this purpose, we stained expanded human U2OS cells with WGA (Wheat Germ Agglutinin), a marker of the nuclear pore inner ring, and either NUP96 or NUP205, two inner components of the NPC (Fig. 2g, h). Using iU-ExM, we were able to distinguish between the inner and outer parts of the NPC. NUP96 was localized on the nuclear and cytoplasmic outer rings, while NUP205 and WGA decorated the inner part of the NPC (Fig. 2g, h). To calibrate the expansion factor achieved by iU-ExM, we used the previously reported NUP96 diameter of 107 nm in SMLM and cryo-electron microscopy (cryo-EM)[24], resulting in a ± 17-fold expansion. With this calibrated expansion factor, we measured that the NUP96 cytoplasmic and nucleoplasmic rings were approximately 64 nm apart (Fig. 2i, j). To further enhance the imaging of NPCs, we focused on isolated nuclei using the NUP96-GFP cell line (Supplementary Fig. 3). We found that iU-ExM similarly enabled the visualization of the canonical features of NPCs, including the 8-fold symmetry and a measured NUP96 thickness of 52 nm (Supplementary Fig. 3a–f and supplementary movies 1 and 2), consistent with previous data that measured the NPC thickness at 49 nm in SMLM[13], 50 nm in cryo-tomography[25], and 57 nm by modelling the SNAP-tag in the electron microscopy structure[26].

Finally, leveraging the capabilities of iU-ExM, we determined the precise iU-ExM expansion factor of macromolecular assemblies in comparison to larger cellular structures such as the nucleus (Supplementary Fig. 4). Additionally, we aimed to compare these results with those obtained through the pan-ExM protocol. To enable a fair comparison, as the ring-like structure of nuclear pores was not clearly visible using anti-NUP96, anti-GFP, and WGA in pan-ExM as compared to iU-ExM (Supplementary Fig. 4a, b), we implemented our intermediate staining approach with pan-ExM to achieve staining that could be directly compared to that of iU-ExM (Supplementary Fig. 4c). Firstly, through the analysis of the physical size of the NPC following U-ExM, pan-ExM (with intermediate staining), and iU-ExM, we observed expanded pore sizes of 367 nm, 757 nm, and 1672 nm, respectively. This confirms that iU-ExM enables an additional four-fold expansion compared to U-ExM, and also indicates that the expansion of NPCs in iU-ExM is more than double that of pan-ExM (Supplementary Fig. 4d).

We then conducted an analysis to determine the expansion factors of the three different approaches using the nucleus cross section as a reference (Supplementary Fig. 4e). These values were used to calculate the corrected diameters of the nuclear pores (Supplementary Fig. 4f). Since these NPC were stained for NUP96, the expected diameter is 107 nm as previously published[13]. Our findings revealed that both U-ExM and iU-ExM exhibited similar performance, with nuclear pores approximately 83 nm in diameter and 50 nm in thickness (Supplementary Figure 4f, g). On the other hand, pan-ExM resulted in NPC of about 50 nm in diameter and 36 nm in thickness. This suggests that although U-ExM and iU-ExM allow for greater expansion at the molecular level compared to pan-ExM, the overall expansion still appears to be lower than expected when considering the reference diameter of 107 nm. Nonetheless, we used this value to calculate the expansion factor at NPC, yielding a factor of 3.3× for U-ExM, 7X for pan-ExM, and 15.7X for iU-ExM (Supplementary Fig. 4h). This result demonstrates that iU-ExM enables twice the molecular expansion compared to pan-ExM, which explains why the symmetry of NPC is now visible. Lastly, considering that increased expansion is anticipated to lead to volumetric dilution and reduced intensity[4,27], we assessed the signal-to-noise ratio across the three methods in the first expansion gel (Supplementary Fig. 5a, b). Our analysis revealed that U-ExM exhibited the

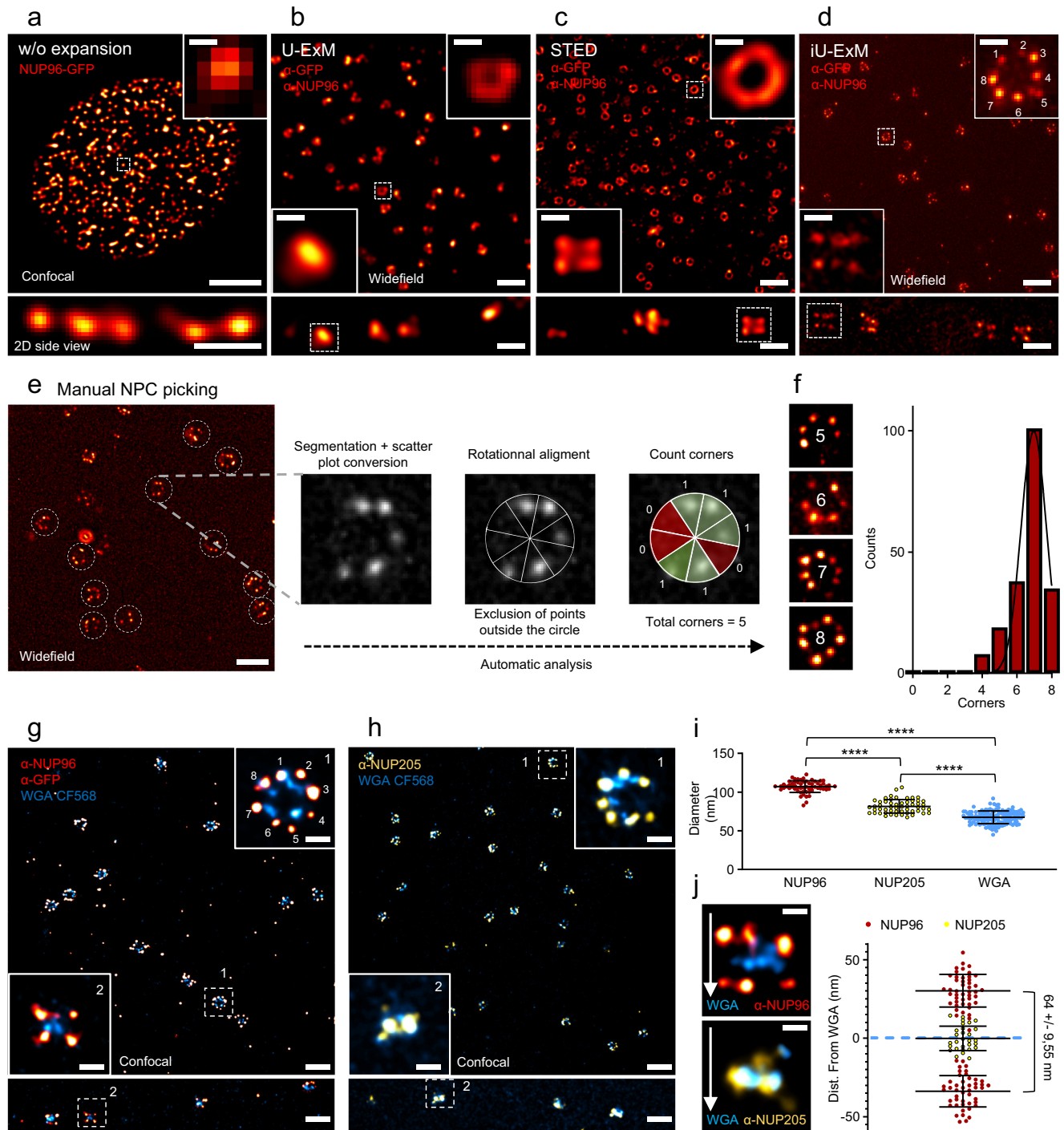

**Fig. 2 | iU-ExM reveals the 8-fold organization of the human Nuclear Pore Complexes. a–d** Upper panel: 2D top view, lower: 2D side view. **a** Confocal image of NUP96-eGFP positive nucleus (red hot) without expansion. Scale bar: 4 µm (upper image), 500 nm (lower), 100 nm (inset). **b** U-ExM widefield image of NPCs stained with α-GFP and α-NUP96 in top view (upper image) and side view (lower image). Scale bar: 500 nm (upper and lower images), 90 nm (inset), corrected. **c** STED images of NPCs stained with α-GFP and α-NUP96 in top view (upper image) and side view (lower image). Scale bar: 500 nm (upper image),150 nm (lower image), 50 nm (inset). **d** iU-ExM widefield picture of NPCs stained with α-GFP and α-NUP96 in top view (upper image) and side view (lower image). Scales bar: 500 nm (upper image), 100 nm (lower image), 50 nm (inset), all corrected. **e** Workflow of the corner counting algorithm. Scale bar (left image): 5 µm non-corrected. **f** Quantification of the corners per NPCs. N = 202 NPCs from 3 independent experiments. Black continuous line: gaussian regression (R² = 0.96).

**g, h** Dual color iU-ExM confocal images NPCs stained for WGA (NPC inner ring, cyan) together with NUP96 (red hot, **g**) or NUP205 (yellow, **h**). Scale bars: 240 nm, insets: 40 nm, corrected. **i** Quantification of the diameter of NUP205, NUP96 and WGA in top viewed NPC. Note that the expanded NUP96 diameter was divided by 107 nm, which represents the published NUP96 value[13] to determine the expansion factor (Exp. F = 16.8×). This value was next applied to the expanded measurements of NUP205 and WGA. NUP96: N = 66 NPCs (average ± standard error = 107 ± 7.2 nm), NUP205: N = 54 NPCs (average ± standard error = 81.6 ± 8.7 nm), WGA: N = 138 NPCs (average ± standard error = 67.3 ± 8.2 nm), from 3 independent experiments. *P*-values: ****$p < 0.0001$. One-way ANOVA. **j** Quantification of the NUP96 and NUP205 position relative to the WGA signal seen in side viewed NPC. NUP96: N = 46 NPCs (average ± standard error = 64 ± 9.5 nm), NUP205: N = 46 NPCs (average ± standard error = 0.5 ± 7.6 nm) from 3 independent experiments. Left images scale bars: 40 nm, corrected.

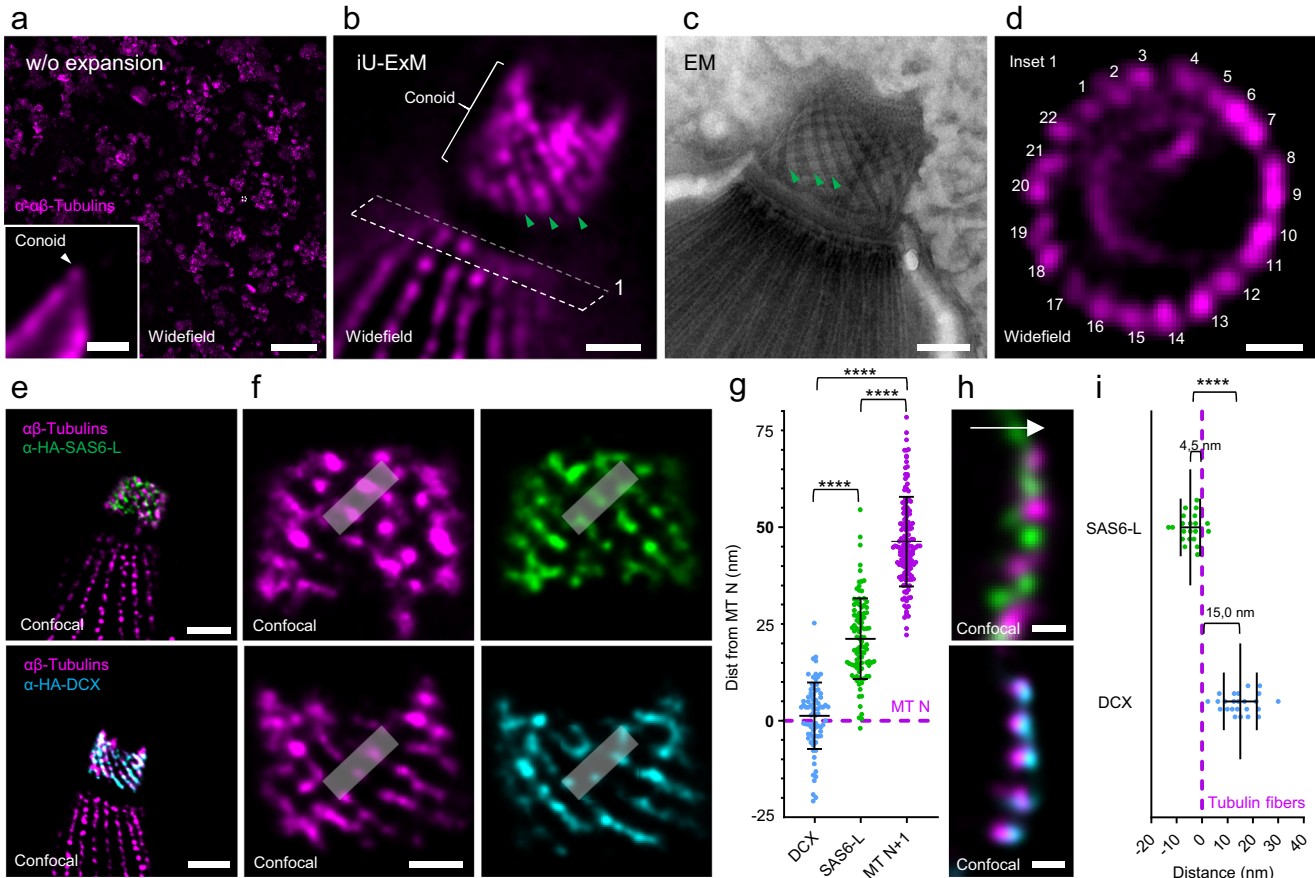

**Fig. 3 | Molecular organization of the conoid in *Toxoplasma gondii* tachyzoites.** **a** Widefield image of non-expanded *T. gondii* tachyzoites labelled with tubulin antibodies (magenta). Inset: one parasite with the conoid region (white arrowhead). Scale bars: 30 μm (full picture), 1 μm (inset). **b** iU-ExM widefield image unveils the spiral shape of the tubulin fibers of the conoid (green arrowheads). Scale bar: 100 nm corrected. **c** Electron microscopy of a conoid showing the tubulin fibers (green arrowheads). Scale bar: 100 nm. **d** iU-ExM widefield image of the region marked by (1) in panel b revealing the 22 cortical microtubules branching the apical polar ring (APR). Scale bar: 100 nm corrected. **e** iU-ExM confocal images of the conoid stained for either HA-tagged SAS6-L (green, upper images) or HA-tagged DCX (cyan, bottom images) and tubulin (magenta). Scales bars: 200 nm corrected. **f** Zoom in on the conoid highlighting the distribution of SAS6-L (green) and DCX (cyan) relative to tubulin (magenta). Scale bars: 80 nm corrected. **g** Quantification of the position of SAS6-L and DCX relative to the tubulin fibers. DCX: N = 80 fibers (average ± standard error = 1.2 ± 8.5 nm), SAS6-L: N = 98 fibers (average ± standard error = 21.1 ± 10.4 nm), MT N + 1: N = 162 fibers (average ± standard error = 46.3 ± 11.6 nm) from 3 independent experiments. **h** Side view of the conoid stained for SAS6-L (green) or DCX (cyan) and tubulin (magenta). Scale bars: 40 nm corrected. **i** Quantification of the position of SAS6-L and DCX relative to the tubulin fibers. DCX: *N* = 24 fibers (average ± standard error = 15.0 ± 6.5 nm), SAS6-L: N = 21 fibers (average ± standard error = −4.6 ± 3.9 nm) from 3 independent experiments. Note that the expansion factor was calculated based on the expanded conoid diameter in the apical part divided by 380 nm[28]. ****p-value < 0.0001. Non-parametric Kruskal Wallis test. (g) and two-sided student t-test (i).

highest signal fluorescence signal, followed by iU-ExM, and finally pan-ExM (with intermediate staining). Along with previous observations on *T. gondii* tachyzoite with the tubulin and DCX staining (Supplementary Fig. 1b), we concluded that this difference in signal-to-noise (S/N) ratio can be explained with the denaturation conditions: a higher temperature also reveals more epitopes and enables higher S/N ratio. In summary, these findings highlight that iU-ExM achieves a molecular expansion of about 16X while maintaining a favorable signal-to-noise ratio for imaging.

### Revealing the molecular architecture of *Toxoplasma* conoid

To further investigate the potential of iU-ExM to reveal the organization of cellular assemblies, we next focused on a tubulin fibers-based structure, the conoid in *T. gondii* (Fig. 3). This structure can serve as a molecular ruler as it displays canonical dimensions of 225–250 nm in length and 380 nm in diameter[28]. *T. gondii* tachyzoites are 4–8 μm long by 2–3 μm wide (Fig. 3a) and reveal their intricate microtubule network and conoid structure from the first round of expansion (Supplementary Fig. 6a, b). In iU-ExM, the cells are significantly larger, resulting in

only a limited number of cells being visible within the field of view (Supplementary Fig. 6c). Moreover, with a conoid with a physical diameter larger than 5 microns in iU-ExM (expansion factor 13.9×, Supplementary Fig. 6d), we found that the organization of the 21 nm apart spiralling tubulin fibers of conoid could be resolved (Fig. 3a–c and Supplementary Fig. 6e–g). This structural feature has until recently only been observed by electron microscopy[29] and lately by cryo-electron tomography[30,31]. In addition, through the comparison of the angles formed by the tubulin fibers with respect to the base of the structure, we observed a consistent angle of 34° as in EM suggesting that the structural organization has been accurately maintained (Supplementary Fig. 6e, f, h). Moreover, we found that the 9-fold symmetry of *T. gondii* centrioles (Supplementary Fig. 6i, j) and the 22 cortical microtubules branching from the apical polar ring (APR)[28] now became clearly resolved (Fig. 3d). However, we found that the two intraconoidal microtubules separated by about 5–10 nm[28] could not be distinguished (Fig. 3d and Supplementary Fig. 6k). Taken together, these findings suggest an effective resolution ranging between 10 nm and 20 nm. This estimation indicates that the theoretical linkage error

of 15 nm, which is expected when using primary and secondary antibodies, may be negligible, as observed in other post-expansion labelling techniques[14]. To validate this, we modelled the effective labelling diameter around a microtubule using either pre-expansion labelling or post-expansion labelling approaches (Supplementary Fig. 6l). It shows that the labelling of microtubules retained the expected linkage error when labelled pre-expansion (55 nm). In contrast, after expansion by 16×, the diameter is expected to be 26.8 nm. Comparing these values with the Full Width Half Maximum (FWHM) measured on the cortical microtubules, we obtained a diameter of 26.7 nm (Supplementary Fig. 6m–p). This result demonstrates that the linkage error diminishes proportionally with the expansion factor and provides evidence that intermediate staining does not exacerbate the linkage error.

Finally, we employed this nanometric resolution to investigate the potential for molecular mapping of the conoid using iU-ExM. Specifically, we examined the spatial distribution of SAS6-L and DCX in relation to tubulin, two proteins previously identified as conoid components[19,20,32] (Fig. 3e–i and Supplementary Fig. 7). We uncovered that SAS6-L is located in between the tubulin fibers, 4.5 nm towards the conoid lumen relative to the fibers, while DCX is located 15 nm from the outer surface of the tubulin signal (Fig. 3e–i and Supplementary Fig. 7d–f), demonstrating the power of resolution of this approach.

### iU-ExM on membrane-based organelles

Imaging membranous organelles in expansion microscopy represents sometimes a significant challenge. For instance, it is possible to use PFA/GA fixation to visualize mitochondria[6], but this may lead to fragmentation of the endoplasmic reticulum (ER)[33]. More generally, the difficulty arises from the fact that some cellular structures are challenging to preserve using the chemical fixatives typically employed in microscopy[34–36]. To address this issue, we have recently introduced a cryo-fixation method that can be combined with U-ExM, called Cryo-ExM[33]. This method has enabled the visualization of membrane structures in their quasi-native state. Notably, we have successfully visualized organelles like the endoplasmic reticulum in human cells or the phase-separated organelle pyrenoid in *Chlamydomonas*[33]. To determine whether iU-ExM was compatible with the visualization of membrane-based organelles and the different types of fixation, we monitored the expansion of mitochondria using PFA/GA fixation and the ER using cryo-fixation (Fig. 4 and Supplementary Fig. 8). We found that iU-ExM expanded and preserved PFA-GA fixed mitochondria stained with NHS ester ATTO 594[12], revealing a cristae spacing of about 85 nm which is within the range of previously reported values[37] (Fig. 4a–c). Similarly, we were able to unveil the tubular organization of the ER by combining iU-ExM with cryo-fixation, achieved through rapid plunging in liquid ethane followed by slow freeze substitution[33] (Fig. 4d, e). Significantly, by adding a small amount of PFA and GA during the freeze substitution step as regularly done in EM preparation (see Methods and Supplementary Fig. 8), we achieved higher precision in imaging the ER, enabling us to visualize the hollow tubules of the ER at a resolution of 45 nm using a confocal microscope (Fig. 4d, e). This level of detail had previously only been observed using super-resolution microscopy techniques such as SMLM[38].

### Ciliary microtubule doublet periodicity revealed in situ

We have demonstrated that iU-ExM offers a resolution comparable to SMLM while providing the benefits of conventional microscopy, such as user-friendliness, fast imaging, and the ability to easily capture multiple colors without the need for specific SRM fluorophores. Next, we undertook to test whether iU-ExM was also compatible with tissue imaging. One of the major limitations of SMLM is its ability to image thick samples such as tissues[3]. To test this, we expanded mouse retinal tissue. The retina is an essential component of the eye responsible for converting light into electrical signals, enabling vision. It consists of photoreceptor cells (Fig. 5a), which have a connecting cilium, a 200 nm

wide cylindrical structure of nine microtubule doublet extending from the centriole/basal body, which facilitates the transport of molecules between the inner and outer segments of the cell[39]. Inside the connecting cilium, we recently revealed using U-ExM the presence of an inner scaffold structure, similar to the one observed inside the centriole, that maintains the microtubule doublets together, partly composed of the Centrin-2 and POC5 proteins[40,41] (Fig. 5b). Around the connecting cilium, all along the microtubule doublets, are found the Y-links, specialized macromolecular structures hypothesized to be made partially of CEP290 (Fig. 5f), a protein crucial for proper photoreceptor function and of which mutations have been linked to eye diseases such Leber Congenital Amaurosis that causes childhood blindness[42–44]. Finally, LCA5, also known as Lebercilin, is also a protein that plays a role in a group of inherited retinal diseases called Leber Congenital Amaurosis. This condition is characterized by the rapid deterioration of photoreceptor cells. In a recent study, we demonstrated that LCA5 is localized above the connecting region of photoreceptor cells, specifically in a region known as the bulge[45]. To challenge our method, we tested iU-ExM and looked at whether we could gain resolution on the position of POC5, Centrin-2, LCA5, and CEP290 along the microtubule doublets of the connecting cilium, directly inside the retinal tissue. To achieve this, we initially embedded the entire retina in the first expansion gel, sectioned it parallel to the photoreceptors axis, stained with antibodies, and continued with the iU-ExM protocol (see "Methods") (Supplementary Fig. 9a–d). During these steps, we first observed with a 20× objective whether the retina was preserved by staining for tubulin and Rhodopsin, which marks the outer segment of photoreceptor cells. We found that cell integrity was maintained and that the connecting cilium and outer segment were clearly expanded with an expansion factor of around 16X (Supplementary Fig. 9e, f). We then repeated the same expansion procedure, this time by staining for tubulin and the inner scaffold component POC5 (Fig. 5a–c and Supplementary Fig. 9g, h).

In the first gel expansion phase, we observed that the signal resembled our previous U-ExM results[40]. POC5 was visible in the centriole and along the connecting cilium (Fig. 5b). In iU-ExM, the signals for POC5 and tubulin appeared dottier, without a distinct pattern (Fig. 5c). However, when we reoriented the gel to obtain cross-sectional views of the connecting cilium and centrioles, we could observe that the signals from tubulin and POC5 did not form a continuous circular pattern as observed in U-ExM (Fig. 5d). Instead, they appeared as nine individual units, likely representing the nine-fold symmetry of the connecting cilium. Interestingly, this nine-fold symmetry of POC5 can also be seen at the level of the centriole/basal body at the base of the connecting cilium as well as in centrioles from cells in culture (Supplementary Fig. 9h, i), suggesting a conserved molecular organization of the inner scaffold. POC5 is found on each microtubule blade, leaning towards the lumen (Fig. 5d and Supplementary Fig. 9g–i). Through the analysis of POC5's position relative to tubulin using a polar transformation followed by a plot profile (Fig. 5e), POC5 clearly does not localize between microtubules but instead aligned perfectly with tubulin suggesting that POC5 is part of a protein complex facing the microtubule doublet. Conversely, we also labelled Centrin, which is known to interact directly with POC5[46], but we were unable to detect any clear 9-fold symmetry (Supplementary Fig. 9j, k). It may be that Centrin follows another pattern inside the inner scaffold that is not specifically along the microtubule doublets. We also investigated the localization of LCA5. In cellular overexpression experiments, this protein has been observed to associate with microtubules, indicating a potential direct interaction. Using iU-ExM, we confirm that LCA5 exhibits a clear localization along the microtubules, specifically in the bulge region, further supporting our previous results obtained in U-ExM[45] (Supplementary Fig. 9l, m).

Finally, we analysed the localization of the Y-link protein CEP290. In U-ExM, CEP290 appears in widefield microscopy as a sleeve covering

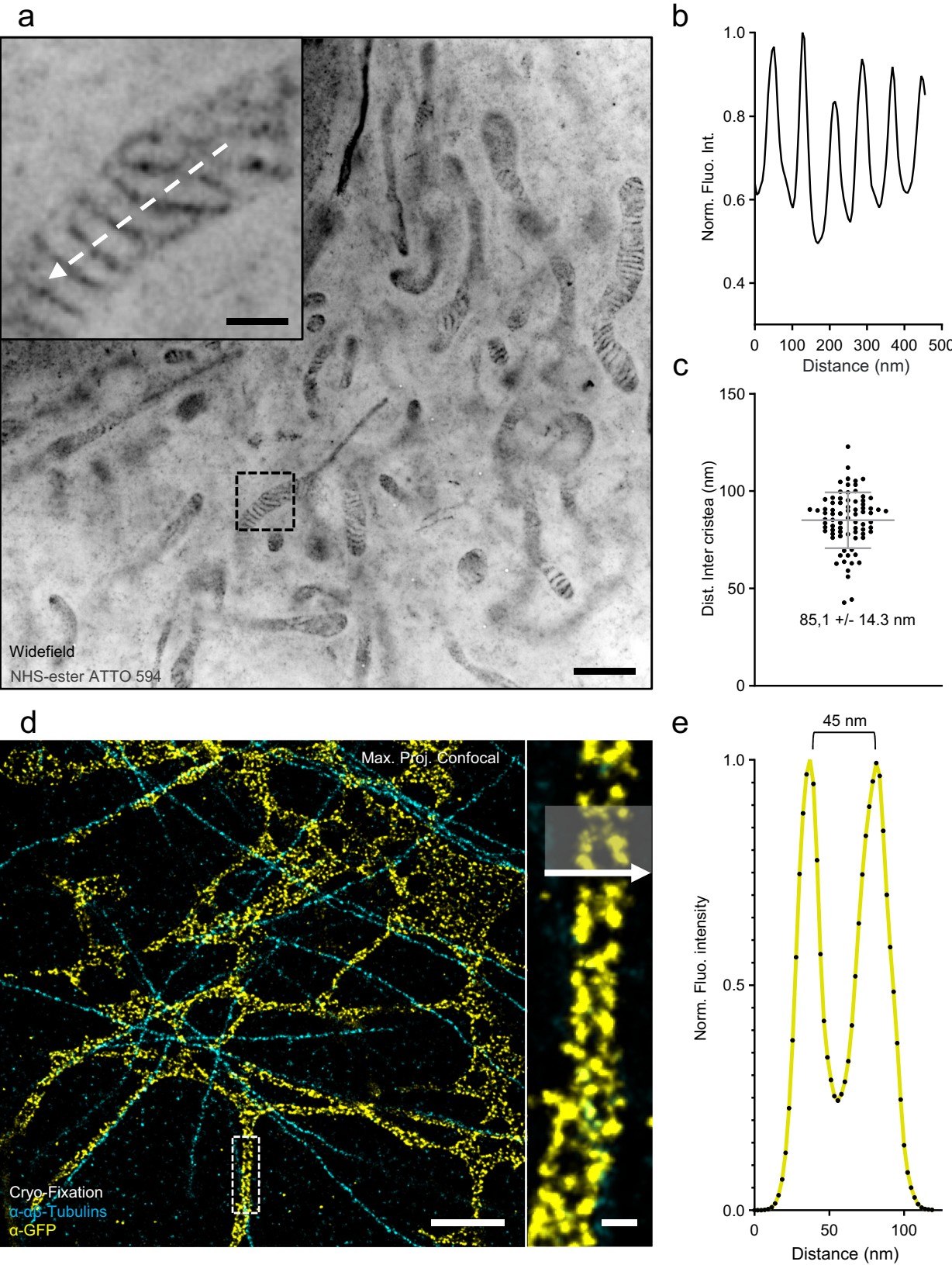

**Fig. 4 | iU-ExM on membranous organelles. a** iU-ExM widefield image of a U2OS cell fixed with 3% PFA, 0.1% GA, and stained with NHS-ester ATTO 594 to label the mitochondria. Note that iU-ExM allows the visualization of individual cristae (inset). Scale bar: 1 μm corrected, 200 nm corrected (inset). **b** Plot profile corresponding to the white dashed arrow in (**a**). **c** Quantification of the inter-cristae distance, N = 78 cristae-to-cristae distances (average ± standard error = 85.1 ± 14.3) nm from 2 independent experiments. **d** iU-ExM confocal image of cryo-fixed U2OS cells expressing Sec61β-GFP and labelled with antibodies against tubulin and GFP, showing the double layer of endoplasmic reticulum tubule. Scale bar: 500 nm (full image), 50 nm (inset), corrected. **e** Plot profile of the grey area shown in (**d**).

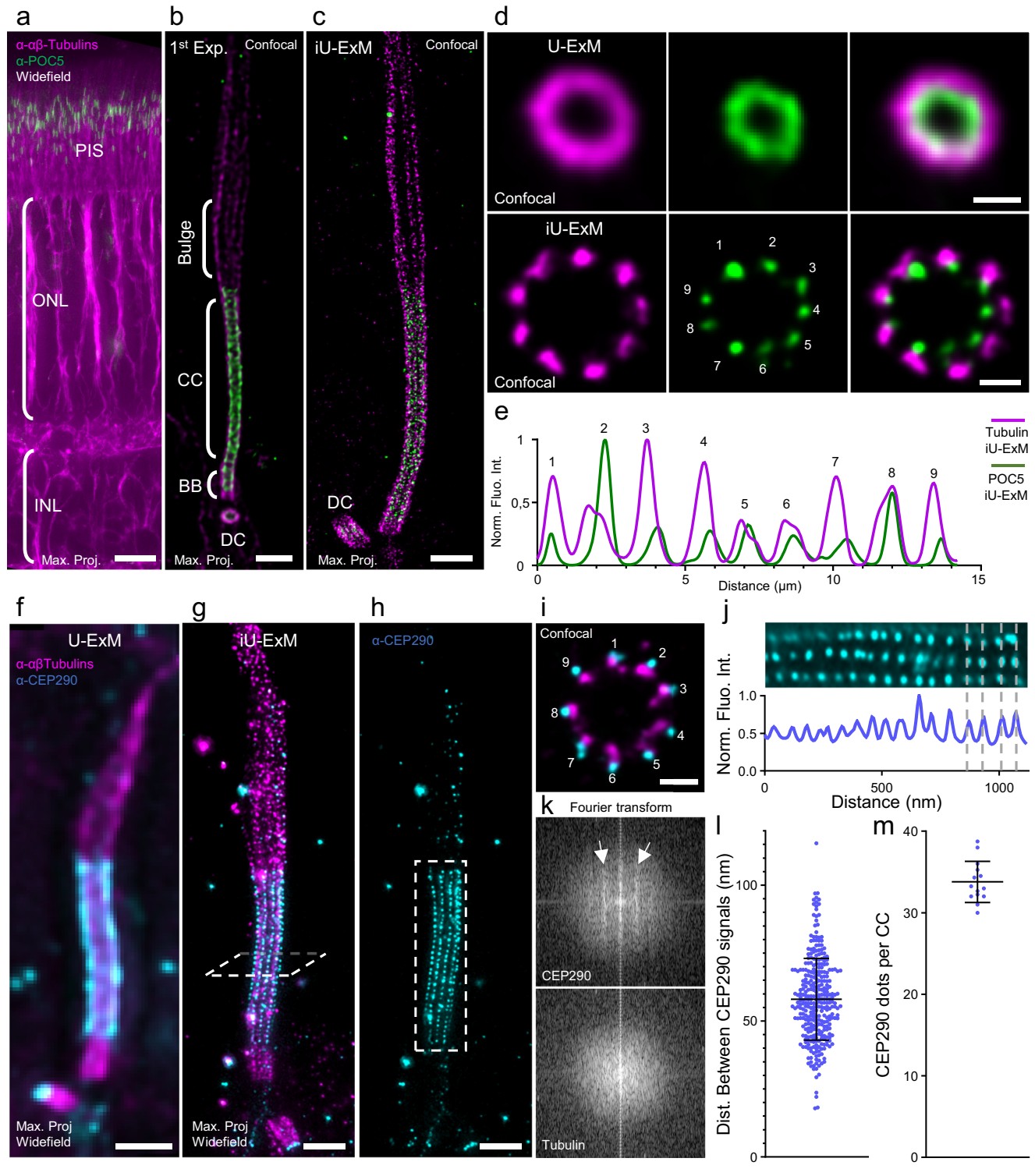

the entire connecting cilium region (Fig. 5f), and in confocal imaging as a nine-fold symmetrical structure (Supplementary Fig. 10a)[40]. Leveraging the enhanced resolution provided by iU-ExM, we found that CEP290 is located on the surface of each microtubule doublet (Fig. 5g–i and Supplementary Fig. 10b). This localisation corresponds to the Y-links, which have been proposed to periodically decorate the microtubule doublets of the connecting cilium[47]. Although we have previously seen that CEP290 forms a linear signal along microtubule doublets in U-ExM[40], we have never been able to detect a periodicity.

However, with the implementation of iU-ExM, we now obtained a remarkably clear CEP290 signal displaying a periodic pattern along each microtubule doublet (Fig. 5g, h). In order to validate the presence of true periodicity in the signal, we conducted a Fourier transform (FFT) analysis on our images. As depicted in Fig. 5j, k and Supplementary Figure 11, the FFT of the CEP290 signal regions revealed distinct layer lines that were not observed in the tubulin signal. This finding provides strong evidence that iU-ExM allows revealing periodic structures along the microtubule doublets. Moreover, we counted the

**Fig. 5 | iU-ExM is compatible with tissue expansion and reveals ciliary periodicity. a** widefield image with a 20× objective of an iU-ExM 1st gel retina tissue stained for tubulin (magenta) and POC5 (green), highlighting the preservation of the different retinal layers. PIS Photoreceptor Inner Segment, ONL Outer Nuclear Layer, INL Inner Nuclear Layer. Scale bar: 25 μm non-corrected. **b** Confocal image of a photoreceptor connecting cilium labelled with tubulin (magenta) and POC5 (green) after the first expansion. CC Connecting Cilium, BB Basal Body, DC daughter centriole. Scale bar: 400 nm corrected. **c** iU-ExM confocal image of a photoreceptor stained for tubulin (magenta) and POC5 (green). Scale bar: 400 nm corrected. **d** Comparison of top view confocal images of the CC expanded using U-ExM (top) or iU-ExM (bottom). Note the 9-fold organization of the microtubule doublets and the POC5 signal revealed after iU-ExM. Scale bars: 100 nm (U-ExM), 50 nm (iU-ExM) corrected. **e** Polar transform plot profile for POC5 (green) and tubulin (magenta) signals from the iU-ExM (continuous) top view images of the CC.

**f** U-ExM widefield image of a connecting cilium stained for tubulin and CEP290. Scale bar: 400 nm, corrected. **g** iU-ExM widefield image of a connecting cilium stained for tubulin and CEP290 revealing ciliary periodicity. Scale bar: 400 nm corrected. **h** CEP290 channel of (**g**). Dashed box: region of interest for Fourier transform. Scale bar identical to (**g**). **i** Confocal top view of the CC with CEP290 and tubulin staining. Scale bar: 50 nm corrected. **j** Plot profile and image of the CEP290 region of the CC showing a periodicity (white arrows). **k** Fourier transform of the region marked by the dashed box in (**h**) demonstrating the periodicity of CEP290 along the CC. **l** Quantification of the periodicity of CEP290 signals along the CC. N = 326 CEP290 dots intervals (average ± standard error= 58 ± 15 nm) from 2 independent experiments. **m** Quantification of the number of CEP290 dots per connecting cilium. N = 14 connecting cilium (average ± standard error= 34 ± 3 CEP290 signals per CC) from 2 experiments.

number of CEP290 dots along each microtubule doublet and found an average periodicity of 57 nm and around 33 dots per microtubule along the CC (Fig. 5i–m), similar to previous electron microscopy studies of the Y-links[47]. We also found an additional localization of the CEP290 at the transition zone, which is the region between the CC and the basal body/centriole (Supplementary Fig. 10c–e). This localization appears different as the CEP290 diameter at the transition zone was larger compared to its diameter at the CC (Supplementary Fig. 10f). This observation suggests that the molecular organization of CEP290 at the transition zone differs from that of the connecting cilium, challenging the previous notion that the connecting cilium is entirely analogous to the transitional zone of motile cilia. Altogether, this demonstrates that iU-ExM enables nanoscale imaging resolution on tissue.

## Enhancing iU-ExM by combining with TREx protocol

The iU-ExM technique enables an approximate 15-fold expansion, derived from the initial expansion of the first DHEBA gel. This gel expands by a factor of around 5.8 when denatured at 85 °C. When reintroduced into the neutral gel, it undergoes slight shrinkage, resulting in an expansion factor of approximately 3.5/4 fold. Subsequently, the second gel expands by a factor of 4–4.5. Multiplying these two expansion factors gives an average final expansion factor of 14–18 times. Currently, it is not feasible to perform a third iteration due to the last gel's inability to cleave. To explore the potential for even greater expansion factors, we decided to combine iU-ExM with the polymers used in the TREx method, which allows a maximum expansion of tenfold[8]. Since TREx employs a gel based on BIS-acrylamide cross-linker, we replaced the third gel in our protocol with the TREx recipe. To evaluate the expansion factor, we used again *Toxoplasma* conoid, *Chlamydomonas* basal body (with defined dimensions, 225 nm wide and 500 nm long[48]), and nuclear pores as molecular rulers (Fig. 6). In all three cases, the combination of iU-ExM with the TREx-based gel resulted in average expansions of approximately 22.4× for the conoid (Fig. 6a, b), 26× for the *Chlamydomonas* basal bodies (Fig. 6c, d), and 22.6× for the nuclear pores (Fig. 6e, f). The average values presented here are based on three replicates, and it should be noted that variations in the expansion factor were observed in each experiment as exemplified in Supplementary Fig. 12. These variations can be primarily attributed to the challenge of accurately controlling the shrinkage of the first gel within the neutral gel. The final expansion factor can be explained by the first gel shrinking at 3.5–4×, which, when multiplied with the 7× TREx-based gel, theoretically gives an expansion factor between 22 and 28×. Furthermore, we did not observe any deformation in the molecular structure, as demonstrated by the organization of the microtubule triplets in the *Chlamydomonas* basal body (Fig. 6c, insets). Overall, this combination of protocols demonstrates that the iU-ExM procedure is compatible with other types of gels. Additionally, the iterative nature of this approach allows for a multiplier

effect on the expansion factors, where even a small difference can result in a significant change in the final outcome.

## Discussion

In this study, we aimed to push the resolution of ExM to the level of SMLM by combining two approaches: U-ExM and iterative expansion microscopy. We optimized each step of the protocol, particularly the homogenization and staining approaches, to achieve an optimal molecular expansion of 16X. The resulting method, called iU-ExM, allowed us now to visualize molecular details.

One of the key aspects of our approach was the use of U-ExM, a variant of the MAP protocol[15], which employed temperature homogenization at 95 °C to preserve the proteome while disrupting protein interactions. However, we found that U-ExM protocol was not readily transferable to the iterative expansion process. Indeed, for the iterative expansion microscopy process, the first gel must be cleaved[11]. This requires the use of DHEBA crosslinker, which in our hands melts during denaturation at 95 °C. To address this issue, we explored the iterative approach outlined in the pan-ExM protocol, which employs DHEBA and denaturation at 73 °C[12]. However, we observed suboptimal immunostaining and expansion outcomes for certain organelles, such as centrioles. We postulated that the temperature disparity between pan-ExM (73 °C) and U-ExM (95 °C) could potentially contribute to these differences. Additionally, the use of a distinct cross-linker in the gel formulation might also impact the immunostaining results. To systematically investigate these factors, we conducted a series of experiments. Our findings revealed that a temperature of 85 °C offered a favourable compromise, as the DHEBA-based gels remained stable at this temperature, resulting in homogeneous expansion and an improved expansion factor in the first gel. We concluded that the homogenization achieved at 73 °C might not be sufficiently robust to ensure complete and uniform expansion. Furthermore, we observed that increasing the temperature also enhanced the intensity and signal-to-noise ratio, likely attributable to improved unmasking of epitopes.

Next, to enhance the outcomes of expansion and immunostaining, we conducted optimizations in the labelling step by incorporating an intermediate staining procedure in the first gel before re-embedding and the application of the second expansion gel. This strategic modification yielded remarkable improvements compare to the pan-ExM protocol, including a substantial increase in the fluorescent signal and enhanced intensity and coverage of labelled structures, such as microtubules. From a technical perspective, the implementation of intermediate staining offers notable advantages, particularly in the context of iterative expansion workflows. The conventional staining conditions established for post-staining in single-round expansion techniques, such as U-ExM, can be directly employed without the requirement for additional optimization in the case of iteratively expanded gels. This streamlines the process and eliminates the need for time-consuming adjustments specific to the post-staining of iteratively expanded gels. Furthermore, the intermediate staining approach

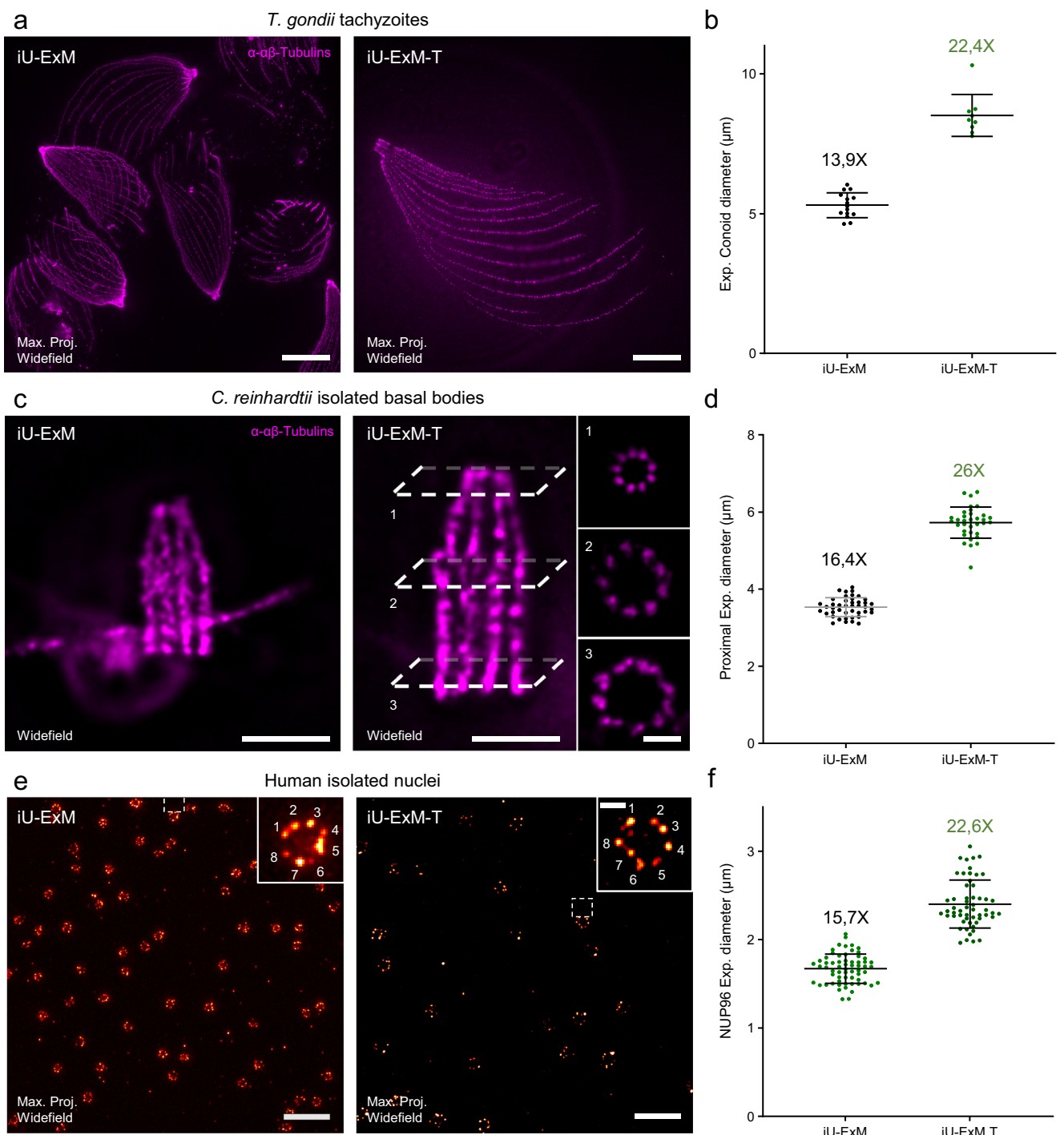

**Fig. 6 | Coupling TREx to iU-ExM. a** Widefield full field of view of *T. gondii* tachyzoites stained for tubulin (magenta) expanded with either iU-ExM (left) or iU-ExM-T (right). Scales bar: 30 µm non-corrected. **b** Quantification of the diameter of the conoid, highlighting the 1.6-fold improvement of the expansion factor. iU-ExM: N = 15 conoids (average ± standard error = 5.3 ± 0.4 µm), iU-ExM-T: N = 9 conoids (M = 8.5 ± 0.7 µm). Data from one experiment. **c** Widefield image of *C. reinhardtii* isolated basal bodies stained for tubulin (magenta) expanded using iU-ExM (right) or iU-ExM-T (left). Scale bars: 5 µm non-corrected. **d** Quantifications of the centriole diameter in the proximal region, further illustrate the 1.6-fold improvement

of the expansion factor. iU-ExM: N = 43 centrioles (average ± standard error = 3.5 ± 0.2 µm), iU-ExM-T: N = 34 centrioles (average ± standard error = 5.7 ± 0.4 µm) centrioles from one experiment. **e** iU-ExM widefield full field of view image of NPCs from isolated U2OS NUP96-GFP nuclei (left) and expanded with iU-ExM-T (right). Scale bars: 10 µm non-corrected. **f** Quantification of the expanded NUP96 diameter between iU-ExM and iU-ExM-T. iU-ExM: N = 55 NPCs (average ± standard error = 1.7 ± 0.2 µm), iU-ExM-T: N = 63 NPCs (average ± standard error = 2.4 ± 0.3 µm) from 3 independent experiments. Expansion factors are calculated by dividing the expanded average diameter by 107 nm.

significantly reduces the volume of the antibody mix necessary for staining the final gel. This reduction in volume is due to the smaller size of the initial gel compared to the iteratively expanded gels. Consequently, the utilization of intermediate staining not only enhances the

efficiency of the technique but also leads to substantial cost savings, a crucial consideration given the typically high cost of antibodies.

In order to evaluate the achieved resolution of iU-ExM, we then directed our attention towards visualizing the nuclear pore complex

(NPC), a macromolecular assembly characterized by 8-fold symmetry, as well as measuring the thickness of microtubules and the conoid of *Toxoplasma gondii* tachyzoites. Through widefield microscopy, we successfully visualized the 8-fold symmetry of nuclear pores by labelling NUP-96 using iU-ExM. Considering that the average distance between neighbouring NUP-96 molecules is approximately 42 nm[13], this shows that the obtained resolution surpasses 40 nm. In fact, we provide evidence that tubulin fibres within the conoid can be visualized, revealing a separation of approximately 21 nm, thus indicating a resolution better than 20 nm. However, we were unable to visualize the individual intra-conoid microtubules, which are spaced only 5 nm apart[28]. Collectively, these results indicate an effective resolution greater than 20 nm but less than 10 nm.

Another crucial factor influencing resolution is the presence of linkage errors, which, in our case, can be attributed to the size of the primary and secondary antibodies. It is worth noting that post-expansion labelling techniques exhibit a reduction in linkage error proportional to the expansion factor[14]. Importantly, in iU-ExM, we observed a decrease in linkage error, even with intermediate staining, resulting in a negligible linkage error when measuring the thickness of microtubules, which was found to be around 26 nm. The achieved final resolution of iU-ExM is therefore comparable to that of super-resolution single-molecule localization microscopy (SMLM) methods. Interestingly, a recent study suggests a promising approach for obtaining even higher resolution than SMLM, by combining expansion microscopy with fluorescence fluctuation analysis, known as ONE (One step Nanoscale)[10]. This innovative technique enables the visualization of protein shapes. Notably, iU-ExM is fully compatible with fluorescence fluctuation analysis, presenting an opportunity to potentially double the resolution attained by ONE, which currently utilizes a 10X expansion factor.

Nevertheless, when achieving such high resolutions, it becomes necessary to consider the potential nanometric deformations that may arise in expansion microscopy. In our investigation, we observed a homogeneous expansion; however, we potentially identified a variation in the expansion factor between the nucleus and the nuclear pores. This discrepancy suggests that certain structures may not undergo complete expansion, which can introduce distortions in measurements even if the expansion is isotropic[49]. The underlying cause for the disparity in expansion factors remains unclear. It is plausible that the expansion factor itself is consistent, but the molecular structure may be subtly affected during the process of chemical fixation or anchoring step, leading to shrinkage and subsequent smaller appearance after expansion. To ensure accurate molecular expansion, it will be crucial in future studies to develop fiduciary markers that enable the measurement of the expansion factor at different scales. These markers will play a critical role in validating and ensuring the correctness of molecular expansion during expansion microscopy. However, it is also important to acknowledge that measurement discrepancies also exist in SMLM. For instance, the dimensions between the cytoplasmic ring and the nuclear ring of the nuclear pore exhibit variations depending on the method employed. SMLM measurements indicate values close to 50 nm[13], whereas the distance between the rings in the EM model measures 57 nm[26]. Once again, chemical fixation likely contributes to the shrinkage of certain molecular structures. One way of circumventing this limitation would be to use an approach we have recently developed, namely cryo-fixation, which can be coupled with expansion microscopy[33]. We tested this approach to visualize nuclear pores in iU-ExM but unfortunately nuclear pores were poorly stained with our antibodies. We also tested coupling chemical fixation and extraction, with cryofixation followed by freeze substitution, but obtained a poor labeling without ring-like structures visible. We conclude that cryofixation coupled with antibody labelling is not optimal for visualizing nuclear pores and that further experiments are needed to improve epitope labelling.

This highlights the need for alternative strategies to antibody-based labelling, such as nanobodies or probes that can be applied post-expansion.

Finally, our study has demonstrated the compatibility of iU-ExM with various gel chemistries by employing the TREx polymer recipe for the final expansion gel. The integration of our intermediate staining approach enables convenient adaptation to different gel chemistries during the final expansion stage and does not necessitate particular staining optimization conditions. These collective findings unveil promising avenues for achieving higher expansion factors and advancing the technique. However, as we strive to achieve expansion factors exceeding 20-fold, potential new limitations may emerge. These limitations encompass the dilution of the signal in three-dimensional space, with a 26-fold expansion corresponding to 17,500-fold increase in volume. Additionally, constraints arise from the working distance of objectives, imposing challenges in capturing detailed structures over greater expansion factors. Furthermore, when targeting specific regions of interest (ROI) within a tissue, such as the small photoreceptor area of the retina relative to the overall tissue size, the identification of suitable ROIs remains a significant challenge.

Overall, our study demonstrates the power of iU-ExM in achieving resolution comparable to SMLM using a conventional fluorescence microscope. By combining U-ExM and iterative expansion, we were able to reveal molecular details. This approach opens up new possibilities for studying subcellular organization and molecular architecture in a broader range of laboratories without the need for specialized equipment. The iU-ExM method can be extended to many cellular structures and tissues, as demonstrated by our successful visualization of the molecular architecture of murine photoreceptors. This advancement in super-resolution microscopy techniques will contribute to a better understanding of cellular processes and aid in various fields of biological research.

## Methods

### Ethical statement
The research performed in this study complies with ethical regulations (authorization VD1367 to the Kostic and Arsenijevic laboratories (Hospital Jules Gonin, Lausanne, Switzerland) regarding mouse retinas).

### Organelles, cell lines, strains and tissue samples
**Chlamydomonas reinhardtii**. To ensure an efficient expansion, we used a cell wall-free *C. reinhardtii* CW15⁻ strain. The liquid culture (TAP buffer[50]) is inoculated from the solid agarose culture of the algae and grown for 3 days at RT under light exposure and slow shaking. The cells were sedimented on 12 mm Poly-D-Lysine coated coverslips for 10 min, then the excess was removed, and the cells were fixed with either cold methanol (see cell fixations) or no fixation.

The purified *Chlamydomonas* basal bodies were prepared and spun on coverslips as previously described[50]. Briefly, deflagellated CW15- *Chlamydomonas* cells were lysed 1 h at 4 °C in presence of 1 mM HEPES (pH 7), 0.5 mM $MgCl_2$, 1% NP-40, and 5000 units of DNase. After centrifugation at $600 \times g$ for 10 min at 4 °C to remove the cell debris, basal bodies were further purified and concentrated, first using a centrifugation at $10,000 \times g$ with a 60% sucrose cushion and second a centrifugation at $68,320 \times g$ on a 40–70% sucrose gradient. Isolated basal bodies were collected at the 70% sucrose interface.

**Toxoplasma gondii**. *T. gondii* tachyzoites were amplified in human foreskin fibroblasts (HFFs, ATCC-CRL-2429, CCD-1112Sk, Lot/Batch No: 70014723) in Dulbecco's Modified Eagle's Medium (DMEM, Gibco) supplemented with 5% of Fetal Bovine Serum (FCS, Gibco), 2 mM glutamine and 25 μg/ml gentamicin (Gibco). Tachyzoites were transfected by electroporation[51]. To target any gene of interest, 40 μg of

specific plasmid driving the expression of the gRNA and Cas9 protein was transfected alongside a PCR product flanked by homology regions (gRNA sequences: DCX: GTGGGGAGCGTGTCACTCAT, SAS6-L: ACT-TATGTACGAGTGCACGG). Transfected parasites then carrying an HXGPRT cassette[52] were selected with 25 mg/ml of mycophenolic acid and 50 mg/ml of xanthine. In brief, SAS6L-mAiD-3HA (HXGPRT) and DCX-mAiD-3HA have been generated by transfecting a Cas9-gRNA encoding vector alongside a PCR fragment encoding for mAiD-HA and HXGPRT cassette. For immuno-staining as well as for U-ExM and iU-ExM, freshly egressed tachyzoites were sedimented for 10 min on pre-coated Poly-L-Lysine coverslips before starting the protocols described in this study. Note that the *T. gondii* tachyzoites used in the paper have the following genetic background: RHΔhxgprtΔKu80 - Tir1 (Parental line for all -mAiD-HA strains generated in this study).

**Human cell lines.** *Homo sapiens* bone osteosarcoma U2OS ATCC-HTB-96 and NUP96-GFP U2OS cell line (from Jonas Ries lab[13]) were grown in Dulbecco's modified Eagle's medium and GlutaMAX, supplemented with 10% fetal calf serum and penicillin and streptomycin (100 μg/ml) at 37 °C in a humidified 5% $CO_2$ incubator. U2OS expressing GFP-sec61β (133−291) were transiently transfected with JetPRIME following the manufacturer's instructions. After 24 h of expression, cells were cryo-fixed as described in ref. 33. All cell cultures were regularly tested for mycoplasma contaminations.

**Mouse retina.** Retinas from Adult mice (2 months) C57BL/6J were dissected as previously described[40]. Two eyes were used from two different mice. Briefly, after enucleation, eyes were directly incubated in 4% PFA (paraformaldehyde, P6148, Sigma-Aldrich) in PBS for 15 min at RT. Then, the eyes were put in a Matek dish (P35G-1.5-10-C, MatTek) for their dissection where the cornea and lens were cut and removed, and the sclera was carefully detached from the retina and discarded. From here, retinas were then directly processed for expansion microscopy.

**Isolated nuclei.** To prepare isolated nuclei[53], cells were harvested from a T75 flask with trypsin and centrifuged at $200 \times g$ for 5 min. The pellet was resuspended in 5 mL of complete DMEM and aliquoted with 1 mL in 1.5 mL Eppendorf tubes. Cells were pelleted with $200 \times g$ for 5 min and resuspended in hypotonic buffer (20 mM Tris-HCl pH 7.4, 10 mM KCl, 2 mM MgCl₂, 1 mM EGTA, 0.5 mM DTT, 0.5 mM PMSF) for 3 min on ice. Then, NP-40 was added to a final concentration of 0.3% for 3 min on ice to lyse the cells with vortexing. The suspension was next centrifuged at 200 g for 5 min and the supernatant (cytoplasmic material) removed. The pellet (nuclei) was resuspended in isotonic buffer (20 mM Tris-HCl pH 7.4, 150 mM KCl, 2 mM MgCl₂, 1 mM EGTA, 0.5 mM DTT, 0.5 mM PMSF) supplemented with NP-40 to reach 0.3% final concentration for 3 min on ice with vortexing to further purify the nuclei. Nuclei were then pelleted at 200 g for 5 min and resuspended in 1× PBS. Finally, purified nuclei were pelleted on 12 mm Poly-D-lysine coverslips at $250 \times g$ for 15 min, by adding one 12 mm coverslip on a well with 500 μL of nuclei preparation. The nuclei were then fixed with 2.4% FA in 1× PBS for 20 min at RT and the coverslips were either expanded or mounted for regular immunofluorescence.

**Cell fixations**
According to the type of structure that we investigated, different fixations were used:

**Glutaraldehyde-formaldehyde fixation.** Immediately after removing the plate from the incubator, the medium was removed from the well and 4 mL of fixing solution (0.1% Glutaraldehyde (GA); 3% formaldehyde (FA) in PBS 1×) was poured for 15 min at RT or at 37 °C (for the mitochondria) in the well without any rinsing to ensure proper fixation. Next, cells were rinsed 3 times with PBS and expanded shortly after.

**Cold methanol fixation.** The coverslips with cells were plunged into cold methanol (−20 °C) and incubated in the freezer (−20 °C) for 7 min.

**Cryo-fixation.** U2OS cells were cryo-fixed, and freeze substituted as previously described[33], with some minor modifications. Briefly, the coverslips containing the sample were held halfway with a thin tweezer (Dumont 5, Sigma F6521-1EA). After blotting the remaining medium, the coverslips were plunged with a homemade plunge freezer in an ethane/propane mix cooled with liquid nitrogen. Coverslips were then transferred into a 5-ml Eppendorf tube containing 1.5 ml of liquid nitrogen-chilled acetone supplemented with paraformaldehyde glutaraldehyde (PFA-GA) at 0.5−0.02% respectively. Tubes were placed on dry ice with a 45° angle and agitated overnight to allow the temperature to rise to −80 °C and further incubated without dry ice for 1 h until the temperature reached ~0 °C. Samples were then rehydrated in successive ethanol: water solutions supplemented with PFA-GA (0.5−0.02%), as follows: ethanol 100%, ethanol 100%, ethanol 95%, ethanol 95%, ethanol 70%, ethanol 50%, ethanol 25% and PBS. Note that coverslips were incubated in the ethanol 100% solutions for 5 min and 3 min for the following. Cells were directly processed for expansion.

**Fixation to visualize nuclear pores.** Cells were pre-extracted as follows[13]. First cells were pre-fixed with 2.4% FA in PBS for 30 s and permeabilized for 3 min with 0.4% Triton X-100 in 1× PBS and washed 2 times with PBS for 5 min. Then, cells were fixed with 2.4% FA in PBS for 20 min at RT and washed 2 times for 5 min with 1× PBS. Finally, a second permeabilization with 0.2% Triton X100 in 1× PBS was performed for 10 min and followed by 2 washes of 1× PBS for 5 min. The coverslips were then stored in 1× PBS before expansion.

**Iterative expansion**
**iU-ExM protocol.** Before starting, a gelation chamber was prepared: on a glass microscope slide, two stacks of two 22 × 22 mm coverslips (no. 1.5) were glued to the slide with enough space for a 12 mm round coverslip in between. Next, one 22 × 22 mm coverslip was added as a lid and secured with 2 coverslips on the side and two on top to create a rack where the lid coverslip could slide in. In this configuration, the lid coverslip cannot move, and the gel thickness would be approximately 170 μm (Supplementary Fig. 13).
    Then proceed to the first expansion following the described steps:
(1)    *Fixation*: Dependent on the sample or imaged organelle. See cell fixations.
(2)    *Anchoring:* The sample on coverslip was incubated in the anchoring solution (2% AA; 1.4% FA in 1× PBS) for 3 h at 37 °C.
(3)    *Gelation*: After anchoring, the excess of the anchoring solution was removed using Kimwipes and the coverslip was sealed in the gelation chamber (Supplementary Fig. 13). The gelation chamber is next put on humid chamber on ice. Using a gelation chamber is important to ensure a controlled and homogeneous height of the gel (around 170 μm). Next, a monomer solution (MS) (10% AA, 19% Sodium Acrylate (SA), 0.1% DHEBA, 0.25% tetramethylethylenediamine (TEMED)/Ammonium Persulfate (APS)) was added to fill the space between the coverslip and the lid of the gelation chamber so that it covers entirely the 12 mm coverslip. After 15 min on ice, the humid chamber was placed at 37 °C for 45 min to complete the gelation.
(4)    *Denaturation*: After gelation, the coverslip with the gel was carefully removed from the imaging chamber and dipped in 2 mL of denaturation buffer (200 mM Sodium Dodecyl Sulfate (SDS); 200 mM NaCl; 50 mM Tris-BASE; pH = 6.8) in a 6-well plate under shaking until the gel detaches from the coverslip. Next, the gel was transferred in a 1.5 mL Eppendorf tube with 1 mL of fresh denaturation buffer and incubated for 1 h30 at 85 °C. The temperature was carefully watched with an external

**Table 1 | Reagents used in this study**

| Product | Supplier | Reference |
| --- | --- | --- |
| N,N′-(1,2-Dihydroxyethylene)-bisacrylamide (DHEBA) | Merck–Sigma Aldrich | 294381 |
| Bis-acrylamide (BIS) | Merck–Sigma Aldrich | M1533 |
| Acrylamide 40% w/w | Merck–Sigma Aldrich | A4058 |
| Formaldehyde 35-38% | Merck–Sigma Aldrich | F8775 |
| Sodium acrylate | AK Scientific | R624 |
| Sodium Hydroxide | Merck–Sigma Aldrich | 206060010 |
| Ammonium persulfate (APS) | Thermo Fisher | 17874 |
| Tetramethylethylenediamine (TEMED) | Thermo Fisher | 17919 |
| DMEM supplemented with glutamax | Thermo Fisher | 61965-026 |
| Trypsin | Thermo Fisher | 25300-054 |
| Fetal Bovine Serum | Thermo Fisher | 10270 |
| Penicillin-Streptomycin | Thermo Fisher | 15140122 |
| Poly-D-Lysine | Merck–Sigma Aldrich | A38904-01 |
| Sodium Dodecyl Sulfate | Pan Reac Applichem | A7219 |
| Tris-Base | Roth | 2449.3 |
| NP-40 | Merck–Sigma Aldrich | I8896 |
| Tween 20 | Roth | 9127-2 |
| Nuclease Free water | Invitrogen | AM9937 |
| Bovine Serum Albumin (BSA) | Merck–Sigma Aldrich | 10735086001 |
| Twinsil Picodent | Picodent | Twinsil |
| Glutaraldehyde | Merck–Sigma Aldrich | G5882 |
| Paraformaldehyde 16% | Electron Microscopy Science | 15710 |

**Table 2 | Antibodies and dyes used in this study**

| Antibody/Dye | Supplier-Reference | Dilution/Concentration |
| --- | --- | --- |
| α Tubulin monobody | ABCD Antibodies - AA344 | 1:100 (iU-ExM), 1:250 (U-ExM) |
| β Tubulin monobody | ABCD Antibodies - AA345 | 1:100 (iU-ExM), 1:250 (U-ExM) |
| HsPOC5 | Bethyl – A303-341A | 1:200 (iU-ExM & U-ExM) |
| GFP | Torrey Pine-TP401 | 1:200 (iU-ExM), 1:250 (U-ExM) |
| NUP98-96 | ProteinTech-12329-1-AP | 1:200 (iU-ExM) |
| NUP205 | ProteinTech-24439-1-AP | 1:200 (iU-ExM) |
| Rat Anti-HA | Roche– 11 867 423 001 | 1:250 (iU-ExM) |
| Rhodopsin | Thermoscientific-MA5-11741 | 1:250 (iU-ExM) |
| WGA CF568 | Biotium-29077-1 | 20 µg/mL (iU-ExM) |
| NHS-Ester | Atto-Tec-AD594-31 | 20 µg/mL(iU-ExM/U-ExM) |
| Rat 488 | Invitrogen – A21208 | 1:250 (iU-ExM), 1:400 (U-ExM) |
| Rabbit 488 | Invitrogen-A11008 | 1:250 (iU-ExM), 1:400 (U-ExM) |
| Mouse 488 | Invitrogen-A11029 | 1:250 (iU-ExM), 1:400 (U-ExM) |
| Mouse 568 | Invitrogen-A11004 | 1:250 (iU-ExM), 1:400 (U-ExM) |
| Guinea Pig 568 | Invitrogen-A11075 | 1:250 (iU-ExM), 1:400 (U-ExM) |
| Centrin | Millipore – 04-1624 | 1:250 (iU-ExM) |
| LCA5 | Proteintech – 19333-1-AP | 1:250 (iU-ExM) |
| CEP290 | Proteintech – 22490-1-AP | 1:250 (iU-ExM and U-ExM) |

thermometer. A few degrees below can cause expansion anisotropy while a few degrees higher can cause gel disintegration.

(5) *1st Expansion step*: after denaturation at 85 °C, the gel was dipped into ddH$_2$O in a 12 cm petri dish. The water was changed every 20–30 min until the expansion of the gel plateaus. We observed that the expansion factor is around 5–6×, depending on the crosslinker purity, pH of the denaturation buffer, temperature, and/or time of denaturation. Note that the DHEBA crosslinker is sensible to pH and temperature, and at 85 °C some crosslinkers are cleaved explaining the gain in expansion factor compared to a 73 °C denaturation.

(6) *Intermediate antibody staining*: the immunolabelling is performed after the 1st expansion step (see staining procedures).

Then process to the 2nd expansion following the described steps:

(7) *Neutral gel embedding*: The first expanded gel was cut into approximately 1 cm$^2$ pieces and placed in a 6-well plate on ice. Then, a piece of gel was incubated 3 times 10 min under shaking and on ice, with activated neutral gel (10% AA; 0.05% DHEBA; 0.1% APS/TEMED in ddH2O). Due to the APS salt content, the gel is expected to shrink about 1.5x. The gel was then put on a microscope slide, and the excess of monomer solution was gently removed using Kimwipes and next, the gel was covered by a 22 × 22 mm coverslip and incubated in a humid chamber for 1 h at 37 °C.

(8) *2nd anchoring*: Note that while this step can be avoided in the pan-ExM original protocol[12], it is required for iU-ExM to retain the antibody staining in the 2nd gel. The gel embedded in the neutral gel was incubated in the anchoring solution (1.4% FA/2% AA) for 3–5 h under shaking at 37 °C. The gel was then washed in PBS 1× for 30 min.

(9) *2nd monomer solution embedding*: In a 6-well plate, the gel was washed 3 times for 10 min under shaking and on ice with the 2nd expansion monomer solution (10% AA, 19% SA, 0.1% BIS, 0.1% TEMED/APS) for a ±16× expansion factor. Next, the excess of monomer solution was gently removed using Kimwipes and the gel was covered by a 22 × 22 mm coverslip and incubated in a humid chamber for 1 h at 37 °C.

(10) *Dissolution of the first and neutral gels:* After final polymerization, the entire gel was incubated in 200 mM NaOH solution for 1 h under shaking at RT followed by washes of ±20 min with PBS 1× until the pH drops to 7.

(11) *Final expansion*: The gel was next dipped in ddH$_2$O and the water was changed until the expansion of the gel plateaus.

**iU-ExM coupled to TREx.** For iU-ExM-T, which gives rise to 22-26X of expansion factor, the final monomer solution is replaced with the TREx[8] monomer solution (14.5% AA; 10.5% SA; 0.01%BIS; 0.1% APS/ TEMED in ddH$_2$O). This monomer solution gave in our hands 7–8× of linear expansion factor in single expansion and not 10×. We attribute this 3–2× discrepancy to the amount of crosslinker in the monomer solution: a lower concentration would have given a more expanded but too fragile gel to be compatible with iU-ExM.

**iU-ExM for mouse retina tissue expansion.** The 1st expansion of the sample was processed as described in ref. 40. Briefly, in a 35 mm Mattek dish with 10 mm microwell (P35G-1.5-10-C), the retina was first embedded in anchoring solution (1.4% FA/2% AA) overnight at 37 °C. After that, the tissue was incubated for 45 min in a monomer solution (19% SA; 10% AA; 0,2% DHEBA in 1x PBS) without APS/TEMED to ensure a proper diffusion of the monomer solution in the tissue. Note that the DHEBA

concentration is doubled to strengthen the gel to ensure proper expansion. Then fresh activated MS was added, and a 22 * 22 mm coverslip was placed on top of the Matek well for 45 min on ice followed by 1 h at 37 °C in a humid chamber. Denaturation buffer was next added to the polymerized gel until the gel popped out of the well. Next, the gel was placed on a 1.5 mL Eppendorf with fresh 1 mL of denaturation buffer and incubated at 85 °C for 2 h. The gel was then expanded in ddH$_2$O and sliced as previously described[40]. The 2nd expansion was performed on the slices as described for the regular iU-ExM protocol.

**pan-ExM Protocol.** The pan-ExM protocol was done as described[12]. Briefly, to analyze centrioles, cells were incubated without fixation in 0.7% formaldehyde + 1% acrylamide (w/v) in 1× PBS for 6 h at 37 °C. After washing the cells in PBS, in a gelation chamber (Supplementary Fig. 13) cells were incubated in the monomer solution containing (19% (w/v) sodium acrylate (SA) + 10% acrylamide (AA) (w/v) + 0.1% (w/v) DHEBA (N,N′-(1,2-dihydroxyethylene) bisacrylamide) + 0.25% (v/v) TEMED (N,N,N′,N′-tetramethylethylenediamine) + 0.25% (w/v) Ammonium persulfate (APS) in PBS and incubated for 1 h at 37 °C in a humid chamber to reach complete polymerization. Then, the gel was dipped in 2 mL denaturation buffer under shaking until the gel detached from the coverslip. The gel was next transferred in a 1.5 mL Eppendorf tube with 1 mL of fresh denaturation buffer and incubated for 1 h at 73 °C. Then, the gel was expanded in ddH$_2$O with at least 3 washes, until the expansion of the gel plateaus. The gel should expand 4–4.5× according to the DHEBA purity. Then, the gel was cut into a 1 cm$^2$ piece and embedded in a neutral gel (10% AA; 0.05% DHEBA; 0.05% APS/TEMED in ddH$_2$O). Embedded gels were incubated in a third monomer solution containing (19% (w/v) SA + 10% AA (w/v) + 0.1% (w/v) BIS + 0.05% (v/v) TEMED + 0.05% (w/v) APS in PBS). After polymerization, the first gel containing DHEBA crosslinkers was dissolved by incubating it in 0.2 M NaOH for 1 h. After several washes, the gels were subjected to immunostaining (see immunostaining below), and placed in distilled water for the final expansion procedure.

**Ultrastructure expansion microscopy (U-ExM).** Expansion of unfixed and cryo-fixed cells was performed as previously described[6,33]. Briefly, cells were incubated for 3 h in anchoring solution (2% AA, 1.4% FA in 1× PBS) at 37 °C before gelation in U-ExM monomer solution (10% AA, 19% SA, 0.1% BIS in 1× PBS) containing 0.5% TEMED and APS. Next, cells were incubated for 5 min on ice followed by 1 h at 37 °C and incubated for 1 h30 at 95 °C in denaturation buffer at pH = 9. Gels were washed from the denaturation buffer twice in ddH$_2$O.

**Labelling and immunostainings**
**iU-ExM intermediate staining/U-ExM gels.** After the first expansion, the gels were shrunk in 1× PBS and stained for 3 h at 37 °C in 1× PBS-BSA 2% for both primary and secondary antibodies, both steps followed by 3 washes for 15 min with 1× PBS-Tween 0.1%. For NPCs, the primary antibody staining was done overnight at 4 °C and secondary for 6 h at 37 °C, both steps followed by 3 washes for 15 min with 1× PBS-Tween 0.1% (see Tables 1 and 2 for concentrations and antibody reference). The gel was next re-expanded in ddH$_2$O.

**iU-ExM/pan-ExM post staining.** After the last expansion in the iterative protocols, the gel was first shrunk in 1× PBS. The primary antibodies were diluted in 1× PBS – BSA 2% (see Tables 1 and 2 for the dilutions). The gel was next incubated with the antibodies for at least 12 h at 37 °C under shaking. The secondary antibodies were diluted in 1× PBS-BSA 2% and incubated with the gel for 6–12 h minimum at 37 °C. Lastly, the gel was washed 3 times for 30 min minimum with 1× PBS-Tween 0.1% and expanded in ddH$_2$O before imaging.

**WGA staining.** For iU-ExM, the WGA staining is done after the antibody labelling on the 1st gel. The gels are shrunk in 1× PBS and incubated under shaking at 37 °C for 1 h30 min with 10 µg/mL of WGA CF-568 in 1× PBS followed by 3 washes for 15 min with 1× PBS, Tween 0.1%.

**NHS-Ester staining.** The final gels (from either U-ExM, iU-ExM or pan-ExM protocols) were incubated 1 h30 at RT under shaking with NHS-Ester ATTO 594 diluted at 20 µg/mL in 1× PBS. The gels were next washed 3 times with PBS-Tween 0.1% for at least 30 min under shaking. Note that the NHS-Ester staining post-expansion can lead to some unspecific signal by additionally labelling the antibodies used for the intermediate staining procedure. For U-ExM/first expanded iU-ExM/pan-ExM gels, the NHS-Ester staining is done after immuno-labelling as NHS-Ester binding might cover the epitopes.

**DAPI staining.** The gel was shrunk in 1× PBS and then stained with DAPI at 1 µg/mL in 1× PBS for 15 min under shaking at RT followed by 3 washes with PBS-Tween 0.1%. Note that for iU-ExM gels, the DAPI staining shows better intensity when labelled with an intermediate staining procedure.

**Gel mounting**
For all stages of expansion (1st or 2nd), gels were cut with a razor blade into squares to fit in a 36 mm metallic imaging chamber. The excess of water was carefully removed using kimwipes, being careful not to dry the gel to avoid shrinking. Next, the gel was placed on a Poly-D-Lysine coated 24 mm coverslips to prevent drifting. Note that the 2$^{nd}$ expanded gels are prone to drift even on Poly-D-Lysine coated coverslips. In case of excessive drift, the gel was stabilized by embedding it in a Twinsil Picodent in the imaging chamber.

**Image acquisition & analysis**
Confocal and widefield images were acquired using either Leica SP8, Leica Stellaris 8, or Leica Thunder, using 63×/1.4 NA oil, 20X/0.4 NA air, 100x/1.47 NA oil objectives. Microscope parameters are controlled using the Suite X software (LAS X; Leica Microsystems). In the figures, if not specifically specified, the images were treated with either LVCC for large images or SVCC for small ROI with widefield images and with Lightning noise reduction for confocal images (Leica Microsystems). Images were processed and quantifications were done with ImageJ (FIJI)[54].

**Gel conservation**
Either stained or non-stained expanded gels can be stored for at least a year in 50% glycerol at −20 °C. For storage, the expanded gel should be washed at minimum 3 times for 1 h with 50% glycerol, until the glycerol completely diffuses in the gel.

To thaw the gels, gels are placed in 1× PBS causing their shrinkage, expelling the glycerol. Next, the gels are washed 2 times for 30 min minimum with PBS1× and then expanded in ddH2O at least once before staining. Insufficient wash would cause weak and noisy antibody labelling for unstained gel and optic aberrations for stained gel due to the altered refractive index due to the presence of glycerol.

**Quantifications**
**Nuclei area measurement.** When the intensity of the DAPI allows it (mostly for 1st expanded gels and non-expanded cells, S/N ratio is often too low with iterative gels), the images were automatically binarized and the particles were selected with the particle tool of ImageJ with a homemade ImageJ macro. For iterative gels, when the DAPI staining did not allow automatic segmentation of the nuclei, areas were manually measured on ImageJ[55]. As the nucleus can be distinguishable with NHS-Ester ATTO 594, Sec61β, or NUP96 staining, the nuclei area could also be manually measured with those staining (at least 10 nuclei).

**Expansion factor measurement.** For iteratively expanded gels, it is difficult to determine the expansion factor by measuring the gel as all the intermediary steps add too many variations. Thus, we relied on

biological rulers. Therefore, we have used ground truth values to measure the expansion factor. For membranous structures (mitochondria, endoplasmic reticulum), the expanded cross section of the nuclei was divided by the non-expanded average cross-section. To obtain the NPC expansion factor, the diameter of the NUP96 signal was measured and divided by 107 nm as previously published[13] to obtain the expansion factor that will be further used for all the other measurements. Between a NUP96 and NUP205 staining of a same experiment, we controlled that the expanded WGA diameter was identical before extending the calculated NUP96 expansion factor to the NUP205 measurements. For *T. gondii*, the expanded diameter of the conoid at the apical part was divided by 380 nm[28]. For *C. reinhardtii* centrioles, the proximal diameter was divided by 225 nm[56]. For retina expansion, the expanded diameter of the basal body at the proximal region was divided by 230 nm, for 50% maximal intensity[40]. Note that we assumed that the linkage error, due to antibody labelling, was neglectable.

**Automatic detection of nuclear pores corners.** To detect the number of corners labelled per nuclear pores, we implemented an adaptation of the algorithm proposed in[13], originally designed only for SMLM data. The input 3D stacks that contain several NPCs are first transformed in 2D max Z-projections. Then individual nuclear pores with ring shape are manually detected and cropped. For each 2D NPC cropped, the number of corners dection was divided in three main steps.

Firstly, the 2D image is transformed into a point cloud, defined as the set of pixel coordinates higher than a given threshold. To prevent the influence of outlier points, we kept only a proportion of the initial number of points (set to 0.9 by default) by discarding the furthest points from the center of the NPC. The set of kept points is denoted $P$.

Secondly, we estimated the orientation of the NPC, in order to align it with the other cropped NPCs. To this end, we applied a rotation to each point of P, to position all the points inside the same sector of the 8-fold symmetry, and we found the average rotation of the points in polar coordinates. This can be formulated as (1)

$$\theta^* = \underset{\theta \in [0, 2\pi]}{\operatorname{argmin}} \sum_{p \in P} \nu\left(\vartheta(p)\left[\frac{2\pi}{s}\right], \theta\right)$$

where $\nu(\theta_1, \theta_2)$ is the angular difference between $\theta_1$ and $\theta_2$, defined by (2)

$$\nu(\theta_1, \theta_2) = (\theta_1 - \theta_2 + \pi)[2\pi] - \pi,$$

$\vartheta(p)$ is the rotation of point $p$ in polar coordinates, and $s$ is the symmetry of the object ($s = 8$ in the particular case of NPC).

Finally, we divided the image in $s = 8$ sectors that separated the corners of the NPC (see Fig. 2e). The sector borders are defined in polar coordinates by (3)

$$B = \left\{\left(R, \theta^* - \frac{\pi}{s} + \frac{2\pi k}{s}\right) k \in [0, s-1]\right\},$$

where $R$ is the distance between the center and the furthest point of $P$. We counted the number of points that belong to each sector, and a sector was said to be activated if it contained a number of points higher than a given threshold, defined as a proportion of the total number of points.

This procedure allowed us to produce the histograms of activated corners Fig. 2 and Supplementary Figs. 2 and 3.

**Statistics and reproducibility**
The normality distribution of every data set was assessed with the Shapiro–Wilk test. If normality passed, ANOVA or student test was runed. If not, non-parametric statistical analysis was done on GraphPad Prism assuming equal SD. All tests were run two-sided. N indicates independent biological replicates from distinct samples. Data are all represented as scatter dot plots with the center line as the mean. The graphs with error bars indicate 1 SD (±) and the significance level is denoted as usual (*$p < 0.05$, **$p < 0.01$, ***$p < 0.001$). All the statistical analyses were performed using Prism7 (GraphPad version 7.0a, April 2, 2016).

All experiments were performed independently at least 3 times, with some exceptions: iU-ExM on mouse retina with CEP290: N = 2. Quantification of the NUP96 dots per NPC for NUP96 antibody only: N = 1 for quantification (N = 3 in total but 2 N where not used for quantification as the fluorescence signal was too weak giving a wrong automatic corner counting). For *T. gondii* micrographs showing the spiral organization of the conoid, micrographs are performed regularly with similar results.

### Reporting summary
Further information on research design is available in the Nature Portfolio Reporting Summary linked to this article.

## Data availability
The datasets generated and/or analysed during the current study are available on Yareta: https://doi.org/10.26037/yareta:wa7d2nbqireshnowx663opofx4. Source data are provided with this paper.

## Code availability
The code for quantifying the number of corners in nuclear pores is available on github: https://github.com/thibaut1998e/NPC_symmetry_quantification/tree/main.

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

## Acknowledgements

We thank Benita Wolf for critical comments on the manuscript and Davide Gambarotto for initiating the project. We thank the Kostic and Arsenijevic laboratories (Hospital Jules Gonin, Lausanne, Switzerland) for providing the mouse retinas used in this study (authorization VD1367), Bohumil Maco for the electron microscopy image shown in Fig. 3, and Nicolas Dos Santos Pacheco who generated the SAS6-L strain *T. gondii* strain. We also thank Yanis Bryois and Sabrina Absalon for their

technical support and advice. This work is supported by the ERC StG 715289 (ACCENT) and the Swiss State Secretariat for Education, Research and Innovation (SERI) under contract number MB22.00075 attributed to P.G., the Swiss National Foundation (SNSF) 310030_205087 and the Pro Visu Foundation attributed to P.G. and V.H., the SNSF 310030_185325 attributed the DSF, and the French National Research Agency (ANR) through the SP-Fluo project (ANR-20-CE45-0007) attributed to D.F.

## Author contributions

V.L. performed all the experiments described in the paper, with the help of R.H. for the expansion of *T. gondii* strains, O.M. with retina, and M.H.L. for cryo-expansion microscopy. R. H. and D.S-F. provided all *T. gondii* strains used in this study. T. E., E.B., and D.F. have developed the method for analyzing the number of corners in nuclear pores images in Fig. 2. V.H. and P.G. conceived, designed and supervised the project. V.L., P.G., and V.H. wrote and revised the final manuscript with inputs from all authors.

## Competing interests

The authors declare no competing interests.
