## [Peer Review File · Nature Communications]

iU-ExM: nanoscopy of organelles and tissues with iterative ultrastructure expansion microscopyREVIEWER COMMENTS

Reviewer #1 (Remarks to the Author):

In their paper “Nanoscopy of organelles and tissues with iterative ultrastructure expansion microscopy (iU-ExM)” Louvel et al. present expansion microscopy images with astonishing resolution. This method is well-documented, shows significant progress, is applied to a variety of samples, and is relevant to the community. However, we identified some issues that they could address before publication:

Major points:

1. It would be interesting to compare the expansion factor at the micrometer and nanometer scale. In the manuscript, different molecular rulers are used to calculate the expansion factor, but only one at a given time. Therefore we suggest, for example, to calculate the expansion factor based on the nuclei cross section and check for the variability of the NPC diameter, and vice versa. This would be informative to judge technical and/or biological variations. Along these lines, it would be informative if the authors discussed the variability of the expansion factor in the main text or in figure legends (e.g., report the standard deviation among the different replicates).
2. When introducing the NPC as reference (from line 76 on), it is important to state that Nup96 is present in 32 copies (each of the 8 corners has 4 copies of Nup96), so the readers can estimate what the presence of e.g. 6 corners in an image of an NPC corresponds to in terms of labeling efficiency. Also, the NPC is not only composed of “two 8-fold symmetrical stacked rings” (lines 77/78), but also contains an inner ring, a nuclear basket, and other subcomplexes.
3. The authors claim that they have quantified the labeling efficiencies (e.g. line 84). However, they quantified the number of corners per NPC and did not calculate the labeling efficiencies based on the counted number of corners as was done by Thevathasan et al. The labeling efficiencies should be calculated during revision, this will also help to quantify the gain in labeling efficiency when staining with anti-Nup96 and anti-GFP antibodies compared to the individual antibodies.
4. Also concerning the counting of corners: The histogram in figure 1C ends at a maximum of 8 dots/NPC, however there are NPCs in the representative image in figure 1B that have more than 8 dots.
5. The analysis of the average angle between Nup96 dots is not very useful to demonstrate the 8-fold symmetry. Angles much smaller than 45° would not be resolved, and angles much larger would not be counted, so there is an intrinsic selection bias. We suggest that instead, the authors could perform an angular autocorrelation analysis (similar to the cross-correlation analysis in Thevathasan et al). An angular autocorrelation curve averaged over all NPCs in the data set would show repeated maxima spaced by 45° if this symmetry is indeed present.

6. The authors find different distances between the nucleo- and cytoplasmic rings for whole cells (64 nm, line 99) and isolated nuclei (52 nm, line 102). This discrepancy should be discussed, also in comparison with other values in the literature:

- SMLM: 49 nm (Thevathasan et al.)

- EM structure: 50 nm (von Appen et al., Thevathasan et al.)

- SNAP-tag modeled in the EM structure : 57 nm (Wang et al., DOI: 10.1101/2022.10.04.510818)

7. The different methods used in the manuscript could be compared and discussed, especially with respect to their advantages and disadvantages. For example, why was the 25x protocol (iU-ExM + TREx) not applied to NPCs? When would the authors choose cryo-fixation over chemical fixation?

Minor points:

1. The method could be discussed more in context. For example, it could be compared to the recently preprinted study on “ONE microscopy” by Shaib et al. (DOI: 10.1101/2022.08.03.502284).

2. It would be interesting to also image NPCs with the cryo-iU-ExM protocol to investigate if the chemical fixation in the pre-extraction method leads to technical variation in, for example, the NPC diameter or non-circular NPCs. However, this is possibly outside the scope of this manuscript.

3. In the figure legends, the authors should state for every scale bar if this is the raw measured value or the value corrected by the expansion factor.

4. For confocal images, it should be stated in the figure legends if and how they were deconvolved.

5. The abbreviation SMLM stands for “single-molecule localization microscopy” instead of “single-molecule light microscopy” (e.g. line 17).

6. The cells were not stained “with either NUP96 or NUP205” (line 92) but with antibodies directed against these proteins.

Similarly in the legend for figure 3 (line 299): The cells were not “labeled with tubulin and GFP” but with antibodies directed against them.

7. The distance between the two NPC rings reported in line 99 should not have ° as unit of the error.

8. Figure 2: The dashed square in panel e that is referred to in f is not visible. What are the shaded squares in f showing?

9. Supplementary Figure 4d: The insert is missing a scale bar.

10. Supplementary Figure 7: Panels a and b are not labeled.

11. Supplementary Figure 8: What is the dashed region in a?

Reviewer #2 (Remarks to the Author):

n/a

Reviewer #3 (Remarks to the Author):

Expansion Microscopy is quickly becoming accepted as a powerful and easy-to-implement method for light microscopy beyond the diffraction limit. The field is rapidly evolving and new or optimized recipes are regularly published, which is important to push the field forward. In this manuscript, Louvel et al describe a new implementation of iterative expansion microscopy. In iterative expansion microscopy, the step of gel expansion is performed twice to facilitate higher expansion factors. This is facilitated by the fact that the first gel is made using reversible crosslinks. Building on their earlier modifications of Expansion Microscopy recipes to preserve the structure of macromolecular complexes in the cell (UExM), the authors now introduce iUExM. They demonstrate a clear improvement in resolution when comparing iUExM and UExM, nicely illustrated using nuclear pore complexes, centrioles, basal bodies, as well as multiple specialized microtubule-based structures found in *Toxoplasma gondii*.

Despite the beautiful images, one important concern that I have is that the iterative approach used by the authors is very similar to an earlier approach introduced by the Bewersdorf lab (M'Saad and Bewersdorf, Nat Comm 2020). Both protocols largely use the same procedure: 1/ Sample gelation in swellable gel (1st gel) with DHEBA crosslinker. 2/ Heat denaturation at pH and temperature that does not degrade DHEBA crosslinker. 3/ Embedding of first gel into a non-swelling gel for stabilization (2nd gel) with DHEBA. 4/ Re-embedding of the second gel in a third, swellable gel with bis as crosslinker. 5/ Incubation in NaOH to dissolve DHEBA crosslinks in 1st and 2nd gels. In addition, both protocols use the same gel recipes, besides the implementation of TReX into iU-ExM in a later part of the manuscript. Nonetheless, the current manuscript introduces some interesting modifications with respect to iterative expansion. For example, it demonstrates that higher denaturation temperatures, up to 85 degrees Celsius, can be applied to the gel without degrading the DHEBA crosslinker. This will in general allow to achieve more homogenous expansion with iterative expansion protocols. Another nice aspect is the combination of different gels into iterative workflows, like U-ExM and TReX. Conceptually however, the paper does not provide real ground-breaking insights and feels somewhat incremental. The iU-ExM protocol mostly mimics pan-ExM, but with slightly optimized denaturation and an additional anchoring step due to labeling after the first expansion step (however, see comment below). The manuscript furthermore demonstrates that iterative expansion is compatible with a range of different fixation strategies (cryo-, methanol, and aldehyde fixation). In addition to this general concern related to overall novelty, I have a number of additional comments that need to be addressed. Overall, I am reluctant to recommend publication in Nature Communications in its present form.

1/ The authors report that intermediate labeling (after 2nd gel but before 3rd gel) is superior to post-expansion labelling with antibodies, but does require additional anchoring of antibodies prior to re-

embedding into the 3rd gel. This adds another 3-5 hours to the protocol will also expose fluorophores to free radicals during gelation, as in pre-expansion staining. To explain why intermediate labeling gives better results, the authors argue that the iU-ExM gels are too dense as result of re-embedding and sterically hinder antibodies from penetrating the gel. However, pan-ExM gels are made from the same recipes and M'Saad et al. (2020) and (2022) already demonstrated that post-expansion labelling with conventional antibodies works well in these gels, both with cells and tissues. This is a confusing aspect of the manuscript that should be clarified.

2/ Following up on the previous point: although the authors describe pan-ExM in the methods section, I could not find a thorough side-by-side comparison of these methods in the manuscript. This is a clear omission that should be addressed.

3/ It would be better if the main figures illustrate the different work flows. For example, Figure S1 would be better as a main figure.

4/ The paper does not end with a satisfying discussion section that compares the approach to earlier work and explains strength and limitations of the current work.

Reviewer #4 (Remarks to the Author):

In this work, the authors used a revised expansion microscopy technique (iU-ExM) to examine the organization of a number of known structures in different organisms, including *Toxoplasma*, which I was asked to provide comments on.

Because of their well-characterized structures, the cortical microtubules and the conoid in *Toxoplasma* were examined to test the iU-ExM technique. The estimated expansion ratio is up to 25-fold, which allows the visualization of fine details in the cortical microtubule array and the conoid that are otherwise elusive using traditional light microscopy. The relative localization of TgDCX and SAS6L with respect to tubulin was investigated, which revealed that DCX and SAS6L are localized to different sides of the conoid fibers under the expansion conditions. Overall, the data quality is high and of interest to the field. However, the following comments need to be addressed.

1) The position of DCX and SAS6L labeling is slightly shifted with respect to that of the tubulin labeling. Anti-tubulin followed by secondary antibodies of two different colors shall be used to determine whether this shift due to the optics (e.g. chromatic aberration) or biology.

2) Related to the first comment, there is no description of how DCX and SAS6L labeling was done. The figure label says " α -SAS6-L" " α -DCX", but SAS6L and DCX antibodies were not listed in the reagent table. The description of the generation of the parasite lines was also very brief. It needs to be much more detailed such that other labs will be able to reproduce the results using the information provided.

3) I am surprised not to see intra-conoid microtubules in these images. They should be easily visible in a cross-section view of the conoid.

4) I assume that all the scale bars for the ExM images were recalibrated based on the estimated expansion ratio. If so, this should be clearly stated in the figure legends.

Reviewer #5 (Remarks to the Author):

In the manuscript titled "Nanoscopy of organelles and tissues with iterative ultrastructure expansion microscopy (iU-ExM) ", Mr. Louvel and colleagues presented the new iterative protocol built on their previous U-ExM method, and demonstrated its performance in studying fine structures in culture cell lines, isolated organelles and unicellular microorganisms. This is an impressive piece with strong presentations of data and analysis, however, there are several issues that need to be addressed before it is ready for publication.

***Major**

1. Overall organization of the paper is rather succinct, in some aspects overly condensed or simplified. First, the abstract will be benefitted from some modest expansion so key features of iU-ExM can be mentioned, and the demonstrated biological examples are sufficiently described. Second, the introduction paragraphs (first two) do not include U-ExM, iterative ExM methods, or pan-ExM whose concepts are so essential to the paper, but instead only one sentence in the third paragraph is used and the authors jumped right into the results. Essential elaborations of why the following biological samples were used in the paper can be very beneficial, too. Further, there are no segmentations (or segment titles) in the results section at all, which is clearly a choice of the authors, but also makes the logic flow of the paper hard to follow. Last, there is no discussion session entirely. Please consider substantially expanding the writing.

2. What is the biggest importance of having iterative U-ExM? This is vaguely answered throughout the paper when results from iU-ExM are presented to be consistent with SMLM or EM measurements, but never clearly stated or summarized. Looking from a different perspective, how much improvement does iU-ExM have over U-ExM? The only explicit comparison seems to be Fig 3i. Elaborating this clearly upfront, with necessary figure rearrangement will greatly help the readers.

3. Relating to the comparisons shown in Fig 3i, it will be valuable to present cross sections of connecting cilium using other markers such as CEP290, LCA5, or GT335, as shown in Mercey et al 2022 (ref [27]), with the polar-transform analysis. These markers have clear, regular, and beautiful periodic organizations that can be revealed by U-ExM, and will foreseeably highlight even better performance from iU-ExM.

4. Core benefit of U-ExM is the minimization of linkage error because pre-expansion staining is avoided

so that antibody hindrance and exaggerated antibody-to-target distance is minimized (as explained in ref [7]). On page 4, line 67-75, the authors described the use of "intermediate staining" which was effectively the pre-expansion staining for the 2nd round of gel, but also reported that it resulted in "a neglectable linkage error" using measurements comparing to previous EM data, without further explanation/discussion. Associated figure panels, Fig S1h, S1i demonstrated the results but could be much stronger if U-ExM vs iU-ExM comparison is provided here.

5. Mouse genotype, dissection procedures, and mounting orientation of the retina tissue were not provided.

*Minor

6. WGA (Wheat Germ Agglutinin) is mentioned throughout without a full name.

7. To avoid confusion, please label "NHS-ester-ATTO594" on images or in legends whenever possible, as "NHS-ester" alone is very misleading, since it is not a fluorochrome in nature, nor an intended target in the sample.

8. Page 3 line 52. "we immunolabeled the centriole using both tubulin and NHS-ester" This is logically wrong. Please revise.

We would like to thank the editor for efficient editorial processing and appreciate the Reviewers' comments and suggestions. We have made all efforts to address the comments and suggestions and detail our replies in the point-by-point response below. Note that as we had to adapt the format of the paper to an article, we modified the entire text and therefore decided not to highlight the changes for the sake of clarity.

REVIEWER COMMENTS

Reviewer #1 (Remarks to the Author):

In their paper "Nanoscopy of organelles and tissues with iterative ultrastructure expansion microscopy (iU-ExM)" Louvel et al. present expansion microscopy images with astonishing resolution. This method is well-documented, shows significant progress, is applied to a variety of samples, and is relevant to the community. However, we identified some issues that they could address before publication:

We thank the reviewer for careful reading of the manuscript and his/her supporting comments and suggestions to further improve the quality of our manuscript.

Major points:

1. It would be interesting to compare the expansion factor at the micrometer and nanometer scale. In the manuscript, different molecular rulers are used to calculate the expansion factor, but only one at a given time. Therefore we suggest, for example, to calculate the expansion factor based on the nuclei cross section and check for the variability of the NPC diameter, and vice versa. This would be informative to judge technical and/or biological variations. Along these lines, it would be informative if the authors discussed the variability of the expansion factor in the main text or in figure legends (e.g., report the standard deviation among the different replicates).

We thank the reviewer for this comment, which we believe improved the quality of the manuscript. We have as suggested calculated the expansion factor based on the nuclei cross section NCS and used it to check for the variability of NPC diameter. This new data can be found in the **revised supplementary Figure 4**. Overall, we found that the expansion factor between the nucleus and the nuclear pores are slightly different, albeit isotropic. By comparing the expansion factors in U-ExM, Pan-ExM, and iU-ExM, we found that U-ExM and iU-ExM give similar results, with an expansion factor at nanoscale close to the micrometer one. However, we found a stronger difference for the Pan-ExM. A possible explanation of this difference might arise from several factors such as the fixation, homogenization efficiency (denaturation), the labeling or the gel expansion itself. We now discuss this discrepancy in the discussion part.

2. When introducing the NPC as reference (from line 76 on), it is important to state that Nup96 is present in 32 copies (each of the 8 corners has 4 copies of Nup96), so the readers can estimate what the presence of e.g. 6 corners in an image of an NPC corresponds to in terms of labeling efficiency.

Also, the NPC is not only composed of "two 8-fold symmetrical stacked rings" (lines 77/78), but also contains an inner ring, a nuclear basket, and other subcomplexes.

We apologize for the lack of clarity regarding the NPC and have adapted the text to include the requested changes. For the simplicity, we describe the NPC as “made **in parts** of two 8-fold symmetrical stacked rings”.

3. The authors claim that they have quantified the labeling efficiencies (e.g. line 84). However, they quantified the number of corners per NPC and did not calculate the labeling efficiencies based on the counted number of corners as was done by Thevathasan et al. The labeling efficiencies should be calculated during revision, this will also help to quantify the gain in labeling efficiency when staining with anti-Nup96 and anti-GFP antibodies compared to the individual antibodies.

We thank the reviewer for asking more quantification of our method. Following Thevathasan et al. approach, that states “Our Nup96 cell lines provide a simple assay to directly measure absolute effective labeling efficiency (ELE). When the ELE is low, NPCs appear as incomplete rings with missing corners. Thus, by statistically analyzing the number of corners of many NPCs, we can infer the absolute ELE”, we used the same cell line, and focused our analysis on the number of detectable corners. However, we had to develop our own algorithm as the public software associated to the published approach was dedicated to localizations coordinates generated by SMLM methods. Therefore, some probabilistic analyses were also not possible. We now provide the corner analysis in the **revised Fig. 2** and revised **Supplementary Fig. 2-3**.

4. Also concerning the counting of corners: The histogram in figure 1C ends at a maximum of 8 dots/NPC, however there are NPCs in the representative image in figure 1B that have more than 8 dots.

As in Thevathasan et al., our methodology defines a maximum of 8 dots per NPC. If a NPC contains more than 8 corners, our algorithm and the one of Thevathasan et al. will assign it only 8 corners. As shown in **Fig. 2f**, the number of corners seems to follow a gaussian distribution centered around 7 (as Thevathasan et al.). If a significant number of 9-fold and 10-fold NPCs were present, the bin corresponding to 8 corners would be artificially increased and the shape of the gaussian distribution will be altered, which suggest these case rarely happen in practice.

5. The analysis of the average angle between Nup96 dots is not very useful to demonstrate the 8-fold symmetry. Angles much smaller than 45° would not be resolved, and angles much larger would not be counted, so there is an intrinsic selection bias. We suggest that instead, the authors could perform an angular autocorrelation analysis (similar to the cross-correlation analysis in Thevathasan et al). An angular autocorrelation curve averaged over all NPCs in the data set would show repeated maxima spaced by 45° if this symmetry is indeed present.

We agree with the reviewer that this analysis was not very useful, so we removed our angles data analysis from this paper. Concerning the suggestion of the reviewer, we have initially tried to perform an angular autocorrelation analysis similar to that of Thevathasan et al. However, the approach is dedicated to localizations coordinates generated by SMLM methods and could not be easily adapted to our data.

6. The authors find different distances between the nucleo- and cytoplasmic rings for whole cells (64 nm, line 99) and isolated nuclei (52 nm, line 102). This discrepancy should be discussed, also in comparison with other values in the literature:

- SMLM: 49 nm (Thevathasan et al.)
- EM structure: 50 nm (von Appen et al., Thevathasan et al.)

- SNAP-tag modeled in the EM structure : 57 nm (Wang et al., DOI: 10.1101/2022.10.04.510818)

We thank the reviewer for these references, we added them in the text and now discuss these values in the discussion part.

7. The different methods used in the manuscript could be compared and discussed, especially with respect to their advantages and disadvantages. For example, why was the 25x protocol (iU-ExM + TReX) not applied to NPCs? When would the authors choose cryo-fixation over chemical fixation?

We thank the reviewer for this comment, which was also requested by another reviewer. To fully enable a clear evaluation of iU-ExM, we now provide a thorough comparison of our new method to the existing Pan ExM as well as U-ExM using 3 molecular rulers: *Toxoplasma gondii* conoid, centrioles and nuclear pore complexes. We also carefully monitored the molecular expansion reached by these 3 methods. The data are now reported in:

- **revised Figure 1:** temperature effect on the expansion in the 3 methods assessing centrioles, conoids and nuclei expansion (1st expansion round).

-**revised supplementary Figure 1:** expansion of *Toxoplasma* conoid using Pan-ExM and iU-ExM (1st expansion gel).

-**revised supplementary Figure 4:** expansion of nuclear pore complexes NPC using Pan-ExM and iU-ExM (final expansion gel). We even, for a fair comparison, introduced our intermediate staining strategy using the Pan-ExM protocol to be able to measure NPC diameter.

Moreover, as requested by the reviewer, we applied iU-ExM combined with TReX to analyze not only the centrioles from *Chlamydomonas* as in the original paper but as well the *Toxoplasma* conoid and NPCs (**revised Figure 6** and **revised Supplementary Figure 11**).

Concerning cryo-fixation versus chemical fixation, it is indeed a critical point in expansion microscopy (but also valid for all methods of super-resolution). In our hands, cryo-fixation works for all approaches but, for example, nuclear pores are better labelled with pre-extraction, approach that requires chemical fixation. We concluded that cryofixation preserves the molecular environment of the nuclear pores so well that the epitopes are hidden. We now discuss this point in the discussion.

Minor points:

1. The method could be discussed more in context. For example, it could be compared to the recently preprinted study on “ONE microscopy” by Shaib et al. (DOI: 10.1101/2022.08.03.502284).

We now added a sentence to compare our method with the ONE in the discussion part.

2. It would be interesting to also image NPCs with the cryo-iU-ExM protocol to investigate if the chemical fixation in the pre-extraction method leads to technical variation in, for example, the

NPC diameter or non-circular NPCs. However, this is possibly outside the scope of this manuscript.

We thank the reviewer for this comment. We initially performed an experiment with the cryo-fixation followed by iU-ExM and looked at the NPCs. However, probably as this method fully preserves all epitopes as well as the crowded cellular environment, the labeling efficiency of the NPCs was extremely low. We therefore undertook to use the well-established pre-extraction protocol to enhance epitope accessibility of the NPCs. This approach led to a very good signal to noise ratio in iU-ExM. To highlight this possible limitation, we also now discuss the fixation in the discussion.

3. In the figure legends, the authors should state for every scale bar if this is the raw measured value or the value corrected by the expansion factor.

For every figure, we have now stated if the scale bar corresponds to the physical (raw) or corrected value by the expansion factor.

4. For confocal images, it should be stated in the figure legends if and how they were deconvolved.

Most of our images are indeed processed post-imaging, not with a deconvolution software but with the “lighting” process from Leica, that corresponds to a noise reduction approach. We included the statement in the material and method section.

5. The abbreviation SMLM stands for “single-molecule localization microscopy” instead of “single-molecule light microscopy” (e.g. line 17).

We apologize for this mistake, we corrected it.

6. The cells were not stained “with either NUP96 or NUP205” (line 92) but with antibodies directed against these proteins.

Similarly in the legend for figure 3 (line 299): The cells were not “labeled with tubulin and GFP” but with antibodies directed against them.

We apologize, we corrected the main text and legend

7. The distance between the two NPC rings reported in line 99 should not have ° as unit of the error.

This error has been corrected.

8. Figure 2: The dashed square in panel e that is referred to in f is not visible. What are the shaded squares in f showing?

This error has been corrected.

9. Supplementary Figure 4d: The insert is missing a scale bar.

We apologize for the omission that we have now corrected.

10. Supplementary Figure 7: Panels a and b are not labeled.

This error has been corrected.

11. Supplementary Figure 8: What is the dashed region in a?

This error has been corrected.

Reviewer #2 (Remarks to the Author):

n/a

Reviewer #3 (Remarks to the Author):

Expansion Microscopy is quickly becoming accepted as a powerful and easy-to-implement method for light microscopy beyond the diffraction limit. The field is rapidly evolving and new or optimized recipes are regularly published, which is important to push the field forward. In this manuscript, Louvel et al describe a new implementation of iterative expansion microscopy. In iterative expansion microscopy, the step of gel expansion is performed twice to facilitate higher expansion factors. This is facilitated by the fact that the first gel is made using reversible crosslinks. Building on their earlier modifications of Expansion Microscopy recipes to preserve the structure of macromolecular complexes in the cell (UExM), the authors now introduce iUExM. They demonstrate a clear improvement in resolution when comparing iUExM and UExM, nicely illustrated using nuclear pore complexes, centrioles, basal bodies, as well as multiple specialized microtubule-based structures found in *Toxoplasma gondii*.

We thank the reviewer for careful reading of the manuscript and his/her comments notably regarding the improvements brought by our new method, which we believe further enhanced the quality of our manuscript. We hope that with these new additions, the reviewer will be convinced that our manuscript deserves publication in *Nature Communications*.

Despite the beautiful images, one important concern that I have is that the iterative approach used by the authors is very similar to an earlier approach introduced by the Bewersdorf lab (M'Saad and Bewersdorf, Nat Comm 2020). Both protocols largely use the same procedure: 1/ Sample gelation in swellable gel (1st gel) with DHEBA crosslinker. 2/ Heat denaturation at pH and temperature that does not degrade DHEBA crosslinker. 3/ Embedding of first gel into a non-swelling gel for stabilization (2nd gel) with DHEBA. 4/ Re-embedding of the second gel in a third, swellable gel with bis as crosslinker. 5/ Incubation in NaOH to dissolve DHEBA crosslinks in 1st and 2nd gels. In addition, both protocols use the same gel recipes, besides the implementation of TREx into iU-ExM in a later part of the manuscript. Nonetheless, the current manuscript introduces some interesting modifications with respect to iterative expansion. For example, it demonstrates that

higher denaturation temperatures, up to 85 degrees Celcius, can be applied to the gel without degrading the DEHBA crosslinker. This will in general allow to achieve more homogenous expansion with iterative expansion protocols. Another nice aspect is the combination of different gels into iterative workflows, like U-ExM and TREx. Conceptionally however, the paper does not provide real ground-breaking insights and feels somewhat incremental. The iU-ExM protocol mostly mimics pan-ExM, but with slightly optimized denaturation and an additional anchoring step due to labeling after the first expansion step (however, see comment below). The manuscript furthermore demonstrates that iterative expansion is compatible with a range of different fixation strategies (cryo-, methanol, and aldehyde fixation). In addition to this general concern related to overall novelty, I have a number of additional comments that need to be addressed. Overall, I am reluctant to recommend publication in Nature Communications in its present form.

While we acknowledge that several iterative expansion microscopy protocols already exist, including Pan-ExM, we are convinced that iU-ExM brings important modifications that allow better expansion of macromolecular complexes as well as a better staining scheme revealing features that would not be seen with Pan-ExM, as explained in detail below.

Moreover, and as requested by the reviewer, to fully enable a clear evaluation of iU-ExM, we provide a thorough comparison of our new method to the existing Pan ExM as well as U-ExM using 3 molecular rulers: *Toxoplasma gondii* conoid, centrioles and nuclear pore complexes. We also carefully monitored the molecular expansion reached by these 3 methods. The data is now reported in:

- **revised Figure 1:** temperature effect on the expansion in the 3 methods assessing centrioles, conoids and nuclei expansion (1st expansion round).

-**revised supplementary Figure 1:** expansion of *Toxoplasma* conoid using Pan ExM and iU-ExM (1st expansion gel). Note that in this figure, we also explain the expansion of the DHEBA gel at 85°C. While the reviewer is correct specifying that at “*higher denaturation temperatures, up to 85 degrees Celcius, can be applied to the gel without degrading the DEHBA crosslinker*”, we wanted to also emphasize that some DHEBA crosslinker get degraded with higher temperature, but not enough to degrade the gel integrity, which results in turns in a higher expansion factor of the first gel (from 4-4.5X to 5.5-6.5X). Moreover, 1X improvement in the first gel leads to 3-4X improvement in the final gel. Altogether, it allows a greater expansion compared to the Pan-ExM denaturation conditions.

-**revised supplementary Figure 4:** expansion of nuclear pore complexes NPC using Pan ExM and iU-ExM (final expansion gel). As in our hands despite multiples trials and staining conditions, we were not able to obtain satisfying labelling of the NPCs. We, therefore, for a fair comparison, introduced our intermediate staining strategy using the Pan-ExM protocol to be able to measure NPC dimensions.

While we understand that the reviewer feels that all these optimizations (temperature, staining) seemed minimal “*The iU-ExM protocol mostly mimics pan-ExM, but with slightly optimized denaturation*”, these are actually crucial to allow a better expansion and to unveil previously unseen expanded molecular assemblies such as NPCs. We believe that these improvements, notably the more homogeneous expansion, a better epitope accessibility and a higher expansion factor, are improving the protocol enough to allow substantial biological discoveries that would not have been accessible with the Pan-ExM protocol. Examples of these discoveries are the 8-fold symmetry of NPCs now visible with a widefield microscopes as well as the CEP290 periodicities visible along the connecting cilium of photoreceptor cells, which had only been visible by electron microscopy (**revised Figure 5, revised Supplementary Figures 9 and 10**).

1/ The authors report that intermediate labeling (after 2nd gel but before 3rd gel) is superior to

post-expansion labelling with antibodies, but does require additional anchoring of antibodies prior to re-embedding into the 3rd gel. This adds another 3-5 hours to the protocol will also expose fluorophores to free radicals during gelation, as in pre-expansion staining. To explain why intermediate labeling gives better results, the authors argue that the iU-ExM gels are too dense as result of re-embedding and sterically hinder antibodies from penetrating the gel. However, pan-ExM gels are made from the same recipes and M'Saad et al. (2020) and (2022) already demonstrated that post-expansion labelling with conventional antibodies works well in these gels, both with cells and tissues. This is a confusing aspect of the manuscript that should be clarified.

We thank the reviewer for this comment, which indeed needs more clarification. To address this point in full, we now include a panel illustrating the difficulties arising with the post-staining strategy using both pan ExM and iU-ExM. We found that in our hands, post staining does not work efficiently after iterative expansion and gives a strong background and an unspecific signal (**revised Figure 1 and revised supplementary Figure 4b**). For this reason, we undertook to develop an alternative staining strategy that we called intermediate staining that allows strong and reproducible staining of the expanded sample after the first expansion (**revised Figure 1m-o**). As the NHS-ester staining works post-expansion, we hypothesized that the poor antibody staining might be due to diffusion of larger molecules inside the final gel.

It is to be noted that this intermediate staining also improves greatly staining in the Pan ExM protocol (**revised supplementary Figure 4b-c**). Moreover, we disagree with the statement that the intermediate staining increases the time of an already long protocol. In contrast, the staining time is reduced to 3h for the primary antibody step and 3h for the secondary antibody incubation, compared to 6h minimum, up to 24h described by M'Saad & al. for the same steps. In addition, it is worth noticing that performing the staining after the first expansion allows reducing the amount of antibody used owing to the size of the 1st gel compared to that of a final gel. This is now discuss at the end of the manuscript.

2/ Following up on the previous point: although the authors describe pan-ExM in the methods section, I could not find a thorough side-by-side comparison of these methods in the manuscript. This is a clear omission that should be addressed.

As stated above, we followed the reviewer recommendation and now provide a side-by-side comparison of the two methods alongside U-ExM using 3 molecular rulers: *Toxoplasma gondii* conoid, centrioles and nuclear pore complexes. These data are now provided in the **revised Figure 1** (centrioles, nuclei and *T. gondii* conoids), **revised supplementary Figure 1** (*T. gondii* conoids), **revised Figure 2** (NPC) and **revised supplementary Figure 4** (NPC). We hope that by this thorough comparison, the reviewer will be convinced on the gain brought in by iU-ExM.

3/ It would be better if the main figures illustrate the different work flows. For example, Figure S1 would be better as a main figure.

We thank the reviewer for this comment, which we followed. The supplementary Figure 1 became now the **revised Figure 1**.

4/ The paper does not end with a satisfying discussion section that compares the approach to earlier work and explains strength and limitations of the current work.

We apologize, indeed the submitted manuscript was written as a report. We now have completely rewritten the paper as an article including a discussion.

Reviewer #4 (Remarks to the Author):

In this work, the authors used a revised expansion microscopy technique (iU-ExM) to examine the organization of a number of known structures in different organisms, including *Toxoplasma*, which I was asked to provide comments on.

Because of their well-characterized structures, the cortical microtubules and the conoid in *Toxoplasma* were examined to test the iU-ExM technique. The estimated expansion ratio is up to 25-fold, which allows the visualization of fine details in the cortical microtubule array and the conoid that are otherwise elusive using traditional light microscopy. The relative localization of TgDCX and SAS6L with respect to tubulin was investigated, which revealed that DCX and SAS6L are localized to different sides of the conoid fibers under the expansion conditions. Overall, the data quality is high and of interest to the field. However, the following comments need to be addressed.

We thank the reviewer for careful reading of the manuscript and his/her supporting comments and suggestions to further improve the quality of our manuscript.

1) The position of DCX and SAS6L labeling is slightly shifted with respect to that of the tubulin labeling. Anti-tubulin followed by secondary antibodies of two different colors shall be used to determine whether this shift due to the optics (e.g. chromatic aberration) or biology.

To unambiguously prove that the DCX and SAS6L labeling relative to tubulin is not due to a shift of colors during microscopic acquisition, we have now included a control where we stained *T. gondii* with two different tubulin antibodies (rabbit anti-tubulin and mouse anti-tubulin), and use secondary antibodies in red and green. We demonstrate that both signals fully co-localize after imaging, indicating that there is no color shift. This data is now reported in the **revised supplementary Figure 6a-c**.

2) Related to the first comment, there is no description of how DCX and SAS6L labeling was done. The figure label says “ α -SAS6-L” “ α -DCX”, but SAS6L and DCX antibodies were not listed in the reagent table. The description of the generation of the parasite lines was also very brief. It needs to be much more detailed such that other labs will be able to reproduce the results using the information provided.

We apologize for this omission or lack of clarity. We now have described in the material and methods section how these lines were generated and adapted the legend of the **revised Figure 3**.

3) I am surprised not to see intra-conoid microtubules in these images. They should be easily visible in a cross-section view of the conoid.

We apologize if the previous version was unclear. Although the intraconoidal microtubules exist, they are not visible in the selected z-sections. We have addressed this by including panels in

Supplementary Figure 5 where the intraconoidal microtubules can be seen. However, due to their proximity of approximately 5-10 nm, our current resolution is insufficient to distinguish between the two microtubules. This resolution limitation is now explicitly stated in the text as well.

4) I assume that all the scale bars for the ExM images were recalibrated based on the estimated expansion ratio.

We apologize for the lack of clarity regarding the scale bars. Indeed, the ExM images were recalibrated based on the expansion factor. This is now stated in the figure legends.

Reviewer #5 (Remarks to the Author):

In the manuscript titled "Nanoscopy of organelles and tissues with iterative ultrastructure expansion microscopy (iU-ExM)", Mr. Louvel and colleagues presented the new iterative protocol built on their previous U-ExM method, and demonstrated its performance in studying fine structures in culture cell lines, isolated organelles and unicellular microorganisms. This is an impressive piece with strong presentations of data and analysis, however, there are several issues that need to be addressed before it is ready for publication.

We thank the reviewer for careful reading of the manuscript and his/her supporting comments and suggestions to further improve the quality of our manuscript.

*Major

1. Overall organization of the paper is rather succinct, in some aspects overly condensed or simplified. First, the abstract will be benefitted from some modest expansion so key features of iU-ExM can be mentioned, and the demonstrated biological examples are sufficiently described. Second, the introduction paragraphs (first two) do not include U-ExM, iterative ExM methods, or pan-ExM whose concepts are so essential to the paper, but instead only one sentence in the third paragraph is used and the authors jumped right into the results. Essential elaborations of why the following biological samples were used in the paper can be very beneficial, too. Further, there are no segmentations (or segment titles) in the results section at all, which is clearly a choice of the authors, but also makes the logic flow of the paper hard to follow. Last, there is no discussion session entirely. Please consider substantially expanding the writing.

We apologize, indeed the submitted manuscript was written as a Brief Communication. We now have completely rewritten the paper in an article format including an introduction, results and a discussion. We hope that the revised manuscript will be easier to follow as such and includes enough introductory material.

2. What is the biggest importance of having iterative U-ExM? This is vaguely answered throughout the paper when results from iU-ExM are presented to be consistent with SMLM or EM measurements, but never clearly stated or summarized. Looking from a different perspective, how much improvement does iU-ExM have over U-ExM? The only explicit comparison seems to be Fig 3i. Elaborating this clearly upfront, with necessary figure rearrangement will greatly help the readers.

We thank the reviewer for this comment, which we have considered. We now provide clear comparison of U-ExM versus iU-ExM in several instances:

-revised Figure 1: first expansion, analysis of nuclei and centrioles.

-revised Figure 2: comparison of U-ExM, STED and iU-ExM for the NPCs. Here, we clearly illustrate the gain in resolution when using iU-ExM, which allows revealing the 8-fold symmetry of NPCs.

-revised supplementary Figure 5: although not exactly a direct comparison of U-ExM and iU-ExM, we compare here the first expansion (panels b and i, similar to U-ExM but with a slightly higher expansion factor) and iU-ExM (2nd expansion, panels c and j).

-revised Figure 5: comparison of the first expansion (U-ExM-like, panel a), U-ExM (panels d, f) and iU-ExM (panels c, g-h) on retinal tissue and photoreceptors. Importantly, the panels f and g show a direct comparison of U-ExM and iU-ExM using a widefield microscope, where the gain in resolution is clearly seen between the two methods.

-revised supplementary Figure 8: comparison of the first expansion (U-ExM-like, panel b), U-ExM (panel e) and iU-ExM (panels d, f) on retinal tissue and photoreceptors.

-revised supplementary Figure 9: comparison of U-ExM (panel a) and iU-ExM (panel b) on retinal photoreceptors.

3. Relating to the comparisons shown in Fig 3i, it will be valuable to present cross sections of connecting cilium using other markers such as CEP290, LCA5, or GT335, as shown in Mercey et al 2022 (ref [27]), with the polar-transform analysis. These markers have clear, regular, and beautiful periodic organizations that can be revealed by U-ExM, and will foreseeably highlight even better performance from iU-ExM.

We thank the reviewer for this comment to further highlight the power of iU-ExM compared to U-ExM. First and as stated above, iU-ExM gives super-resolution images using a regular widefield microscope (**revised Figure 5 f, g**). Second, we have included staining for Centrin, an inner scaffold component although localizing at the connecting cilium and lebercilin (LCA5), which is located at the bulge region. In addition, we stained for CEP290, which marks the outside of the connecting cilium correlating with the presence of the Y-links structure. Interestingly, we found that iU-ExM enables seeing the periodicities of the CEP290 signals along the connecting cilium, which was never seen previously.

4. Core benefit of U-ExM is the minimization of linkage error because pre-expansion staining is avoided so that antibody hindrance and exaggerated antibody-to-target distance is minimized (as explained in ref [7]). On page 4, line 67-75, the authors described the use of "intermediate staining" which was effectively the pre-expansion staining for the 2nd round of gel, but also reported that it resulted in "a neglectable linkage error" using measurements comparing to previous EM data, without **further explanation/discussion**. Associated figure panels, Fig S1h, S1i demonstrated the results but could be much stronger if U-ExM vs iU-ExM comparison is provided here.

We thank the reviewer for this remark on the lack of explanations concerning the statement of a "neglectable linkage error" with intermediate staining. Indeed in the similar way that it has been demonstrated that staining prior to expansion preserves the antibody-to-target distance (linkage

error), the labelling post-expansion leads to a reduction of the linkage error proportional to the expansion factor (DOI: [10.1016/bs.mcb.2020.07.002](https://doi.org/10.1016/bs.mcb.2020.07.002)). In order to find out whether the linkage error was also divided by the expansion factor, despite our intermediate staining, we now first modelled the diameter of the fluorescent signal around the microtubules as a function of the expansion factors. Importantly, we found a perfect match between the simulated data and the measured data, i.e. 26.6nm. This new data is **Supplementary Fig. 5i**.

5. Mouse genotype, dissection procedures, and mounting orientation of the retina tissue were not provided.

We have now included these details in the material and methods section.

*Minor

6. WGA (Wheat Germ Agglutinin) is mentioned throughout without a full name.

We have corrected this in the text.

7. To avoid confusion, please label "NHS-ester-ATTO594" on images or in legends whenever possible, as "NHS-ester" alone is very misleading, since it is not a fluorochrome in nature, nor an intended target in the sample.

We have corrected this along the revised figures.

8. Page 3 line 52. "we immunolabeled the centriole using both tubulin and NHS-ester" This is logically wrong. Please revise.

We apologize and have corrected the sentence as follows: "Centrioles were stained using tubulin antibodies and further detected using the pan NHS ester-ATTO594."

- - - - -

Reviewers' comments:

Reviewer #1 (Remarks to the Author)

In their revised version of the manuscript entitled “Nanoscopy of organelles and tissues with iterative ultrastructure expansion microscopy (iU-ExM)” Louvel et al. improved the initial manuscript substantially and we recommend publication in Nature Communications if our main concern (see below) is answered.

The newly performed experiments to compare the different methods on a variety of biological samples shows the versatility of the approaches and is a valuable resource for the community. The extended introduction and discussion further enhance the manuscript.

Major point:

1. The authors write “We concluded that cryofixation preserves the molecular environment of the nuclear pores so well that the epitopes are hidden”. An alternative interpretation could be that cryofixation destroys more epitopes (e.g. by denaturing caused by acetone etc). We suggest the experiment: Perform cryo-fixation + expansion after chemical fixation and pre-extraction, and then compare the results with cryo-fixation + expansion alone and chemical fixation/pre-extraction + expansion alone.

Minor points:

1. Line 32 still contains a wrong long version of SMLM (“light”). Besides, SMLM (and other abbreviations) are defined multiple times in the text.
2. Line 305: Rephrase “revealing the correct cristae spacing of 85.10 nm” to something like “revealing a cristae spacing of about 85 nm which falls into the range of previously reported values”
3. Fig. 4a: What’s the size of the scale bar in the insert?
4. In SFig. 4f,g,h: The unit should be nm instead of μm .

Reviewer #3 (Remarks to the Author)

The authors have satisfactorily addressed my comments. The figures are now much better organized and show direct comparisons with the pan-ExM protocol. This reveals a clear improvement of their new protocol. I therefore recommend publication

Reviewer #4 (Remarks to the Author)

The authors have addressed all my concerns.

Reviewer #5 (Remarks to the Author)

I want to start off by thanking the authors for providing such a high-quality revision and rebuttal, with which the current manuscript has been improved substantially.

All my major concerns have been properly addressed. The only minor lingering issue is of point #7 in my previous report, regarding the notation of "NHS-ester" where "NHS-ester-ATTO594" is a more accurate and unambiguous mark. All the mentions in the body text and figure legends have been corrected, but figure labels such as the ones in Fig 1k, 1l, 4a were not corrected. If the length of the labels was a concern, marks such as "full proteome", "pan-proteome" or "all proteins", etc. could also be considered. Nevertheless, I still insist on making this point clear as NHS-ester itself is not a natural native component within the cell, nor does it fluoresce.

Other than this, I have nothing further to ask. This is an impressive and exciting piece of work that is guaranteed to accelerate findings on molecular machinery in cells at nanoscopic precision. Well done.

We would like to thank the editor and the reviewers for providing their comments and suggestions for this 2nd round of revisions.

REVIEWER COMMENTS

Reviewer #1 (Remarks to the Author):

In their revised version of the manuscript entitled “Nanoscopy of organelles and tissues with iterative ultrastructure expansion microscopy (iU-ExM)” Louvel et al. improved the initial manuscript substantially and we recommend publication in Nature Communications if our main concern (see below) is answered.

The newly performed experiments to compare the different methods on a variety of biological samples shows the versatility of the approaches and is a valuable resource for the community. The extended introduction and discussion further enhance the manuscript.

We thank the reviewer for his/her supporting comments.

Major point:

1. The authors write "We concluded that cryofixation preserves the molecular environment of the nuclear pores so well that the epitopes are hidden". An alternative interpretation could be that cryo-fixation destroys more epitopes (e.g. by denaturing caused by acetone etc). We suggest the experiment: Perform cryo-fixation + expansion after chemical fixation and pre-extraction, and then compare the results with cryo-fixation + expansion alone and chemical fixation/pre-extraction + expansion alone.

We thank the reviewer for this comment on the previous rebuttal which indeed needed more clarification.

The reviewers suggest that epitopes can be denaturated by the acetone, which we believe is possible. However, the principle of iU-ExM and U-ExM is to perform the immuno-labelling after a heat denaturation step, the antibodies then recognize only denaturated linear epitopes and not native 3D epitopes. We do not think that the epitopes would be more denaturated than the heat denaturation process, coupled with high concentration of chaotropics like Sodium-Dodecyl-Sulfate (SDS).

Also, in the cryo-fixation paper that we published in Nature Methods in 2022 (<https://doi.org/10.1038/s41592-021-01356-4>), by labelling mitochondrial protein TOMM20 in human U2OS cells, we could demonstrate that epitopes are preserved and sometimes better conserved compared to chemical fixations..

To answer reviewer question, we performed some of the experiments requested. We compared the staining of NUP96 on the first step of expansion (figure below) between cells that were pre-extracted + chemical fixation (cond. 1) and pre-extracted + chemical fixation + cryo fixation (cond. 2). While the NPCs can be seen as rings with the cond. 1, they were only seen as dots in the cond. 2 even if the nuclei were similarly expanded (Figure A, B, C). Seeing these results, it appears that chemical fixation, extraction, followed by cryo-fixation as an impact on the NPCs staining. As suggested by the reviewers, the epitopes may be affected by the freeze substitution but it is also possible that the coupling of the two approaches results in excessive extraction. This is very interesting and deserve more investigations to explain this observation.

However, we believe that it goes beyond the line of the paper that is more focused on the development of the iterative approach and not on the optimization of the cryo- fixation method itself.

Figure 1: Effect of Cryo-Fixation on NPC staining. A,B: Widefield image of U2OS-NUP96-GFP cells stained with both α -NUP96 and α -GFP after the 1st expansion step of iU-ExM avec either pre extraction and PFA 2.4% fixation of the cells (see methods) (A) or the same but with cryo-fixation after (B). Scale bar: 2 μ m non corrected. C: Nuclei cross section corrected with the gel expansion factor. Pre E: Pre extraction + PFA 2.4% fixation (N= 93, average +/- SD: 15.7 +/- 2.2 μ m), Pre E + Cryo: Pre Extraction + PFA 2.4% fixation + Cryo-Fixation (N=98, average +/- SD: 15.4 +/- 2.4) from 2 independant experiments. Statistics: non parametric Mann-Whitney test (P-Value=0.147).

For the aforementioned reasons, we have consequently chosen not to incorporate new data that remains unexplainable. However, we agree that our discussion might sound too strong and we propose the following sentences instead“ We tested this approach to visualize nuclear pores in iU-ExM but unfortunately nuclear pores were poorly stained with our antibodies. We also tested coupling chemical fixation and extraction, with cryofixation followed by freeze substitution, but obtained a poor labeling without ring-like structures visible. We conclude that cryofixation coupled with antibody labelling is not optimal for visualizing nuclear pores and that further experiments are needed to improve epitope labelling.”

Minor points:

1. Line 32 still contains a wrong long version of SMLM (“light”). Besides, SMLM (and other abbreviations) are defined multiple times in the text.

We apologies for the lack of consistency regarding this point. We corrected it.

2. Line 305: Rephrase “revealing the correct cristae spacing of 85.10 nm” to something like “revealing a cristae spacing of about 85 nm which falls into the range of previously reported values”

We thank the reviewer for this accurate remark. We corrected the sentence.

3. Fig. 4a: What’s the size of the scale bar in the insert?

We apologies for the lack of consistency regarding this point. We added the value in the figure legend.

4. In SFig. 4f,g,h: The unit should be nm instead of μ m.

We would like to thank the reviewer for spotting this error. We have corrected the figure legend.

Reviewer #3 (Remarks to the Author)

The authors have satisfactorily addressed my comments. The figures are now much better organized and show direct comparisons with the pan-ExM protocol. This reveals a clear improvement of their new protocol. I therefore recommend publication.

We thank the reviewer for his/her very constructive remarks from the first round of revisions, it really improved the manuscript. We are happy that the new version convinced the reviewer.

Reviewer #4 (Remarks to the Author)

The authors have addressed all my concerns.

We thank the reviewer for his/her consideration and are happy that the new version convinced the reviewer.

Reviewer #5 (Remarks to the Author)

I want to start off by thanking the authors for providing such a high-quality revision and rebuttal, with which the current manuscript has been improved substantially.

We thank the reviewer for his/her positive comments.

All my major concerns have been properly addressed. The only minor lingering issue is of point #7 in my previous report, regarding the notation of "NHS-ester" where "NHS-ester-ATTO594" is a more accurate and unambiguous mark. All the mentions in the body text and figure legends have been corrected, but figure labels such as the ones in Fig 1k, 1l, 4a were not corrected. If the length of the labels was a concern, marks such as "full proteome", "pan-proteome" or "all proteins", etc. could also be considered. Nevertheless, I still insist on making this point clear as NHS-ester itself is not a natural native component within the cell, nor does it fluoresce.

We apologies for the lack of consistency regarding the notation of the NHS-Ester Labelling. We added "NHS-ester-ATTO594" on the figures.

Other than this, I have nothing further to ask. This is an impressive and exciting piece of work that is guaranteed to accelerate findings on molecular machinery in cells at nanoscopic precision. Well done.

We thank the reviewers for those kind words.

REVIEWERS' COMMENTS

Reviewer #1 (Remarks to the Author):

In the new revision of the manuscript entitled “Nanoscopy of organelles and tissues with iterative ultrastructure expansion microscopy (iU-ExM)” Louvel et al. answered our main concern and we therefore recommend publication in Nature Communications.

The results of the newly performed experiments are curious but we agree with the authors that further investigation would go beyond the scope of this paper and that the textual change in the discussion is sufficient.

REVIEWERS' COMMENTS

Reviewer #1 (Remarks to the Author):

In the new revision of the manuscript entitled “Nanoscopy of organelles and tissues with iterative ultrastructure expansion microscopy (iU-ExM)” Louvel et al. answered our main concern and we therefore recommend publication in Nature Communications.

The results of the newly performed experiments are curious but we agree with the authors that further investigation would go beyond the scope of this paper and that the textual change in the discussion is sufficient.

We thank the reviewer 1 for agreeing with our revisions and supporting publication of our manuscript in *Nature Communications*.